# EZH2-H3K27me3 mediated KRT14 upregulation promotes TNBC peritoneal metastasis

**Ayushi Verma[1], Akhilesh Singh[1], Manish Pratap Singh[1], Mushtaq Ahmad Nengroo[1], Krishan Kumar Saini[1,2], Saumya Ranjan Satrusal[1,2], Muqtada Ali Khan[1], Priyank Chaturvedi[1], Abhipsa Sinha[1], Sanjeev Meena[1], Anup Kumar Singh[1] & Dipak Datta ®[1,2] ✉**

Triple-Negative Breast Cancer (TNBC) has a poor prognosis and adverse clinical outcomes among all breast cancer subtypes as there is no available targeted therapy. Overexpression of Enhancer of zeste homolog 2 (EZH2) has been shown to correlate with TNBC's poor prognosis, but the contribution of EZH2 catalytic (H3K27me3) versus non-catalytic EZH2 (NC-EZH2) function in TNBC progression remains elusive. We reveal that selective hyper-activation of functional EZH2 (H3K27me3) over NC-EZH2 alters TNBC metastatic landscape and fosters its peritoneal metastasis, particularly splenic. Instead of H3K27me3-mediated repression of gene expression; here, it promotes *KRT14* transcription by attenuating binding of repressor SP1 to its promoter. Further, KRT14 loss significantly reduces TNBC migration, invasion, and peritoneal metastasis. Consistently, human TNBC metastasis displays positive correlation between H3K27me3 and KRT14 levels. Finally, EZH2 knockdown or H3K27me3 inhibition by EPZ6438 reduces TNBC peritoneal metastasis. Altogether, our preclinical findings suggest a rationale for targeting TNBC with EZH2 inhibitors.

According to global cancer statistics, breast cancer is the most common malignancy worldwide in women, having 2.09 million new cases diagnosed in 2018 with 0.6 million deaths[1]. Among different breast cancer subtypes, basal-like breast cancer is often clinically defined as triple-negative breast cancer (TNBC), which lacks immunohistochemistry (IHC) expression of estrogen receptor (ER), progesterone receptor (PR), and human epidermal growth factor receptor 2 (her2/neu) according to College of American Pathologist (CAP)/ASCO guidelines[2]. It involves patients of young age and is associated with high-grade, poor-prognosis tumors[3]. TNBC poses a serious threat to clinicians due to the enormous heterogeneity of the disease and the absence of well-established molecular targets[4]. Indeed, the 90% mortality of breast cancer, including TNBC, is associated with metastasis[5–8]. Underpinning the molecular cues for TNBC metastasis and its therapeutic intervention are the hotspots of current cancer research. The concept of epigenetic reprogramming is admired as a driving force for distant organ metastasis. Importantly, Breast cancer molecular subtypes have been identified with unique chromatin architecture with diverse methylation patterns[9,10]. However, how epigenetic mechanisms exclusively regulate subtype-specific distinct transcriptional nexus still remains elusive. Epigenetic modulator Enhancer of Zeste homolog 2 (EZH2), a catalytic subunit of Polycomb Repressive Complex 2 (PRC2), promotes target genes suppression by tri-methylation of lysine 27 of histone H3 (H3K27me3)[11]. Previously, we have shown the classical gene silencing function of EZH2 where death receptors are epigenetically suppressed in cancer stem cells[12]. Notably, aberrant EZH2 expression has been associated with diverse cancers concerning both oncogenic and tumor suppression functions[13]. Besides its canonical function as a transcriptional repressor, EZH2 protein has recently been shown to perform H3K27me3 independent functions[14–18]. In support of non-canonical functions, studies have shown a hidden, partially disordered transactivation domain (TAD) in EZH2, which directly binds to the

[1]Division of Cancer Biology, CSIR-Central Drug Research Institute (CDRI), Lucknow 226031, India. [2]Academy of Scientific and Innovative Research, Ghaziabad, Uttar Pradesh 201002, India. ✉e-mail: dipak.datta@cdri.res.in

transcriptional coactivator p300 and promotes oncogenesis[17,18]. However, Mittal's group has performed the bioinformatics analysis of epigenetic-associated genes in breast cancer patients, which characterized overexpression of EZH2 catalytic function as a predominant carrier in TNBC metastasis with poor overall survival[19].

Expression of basal cytokeratins like CK5/6, CK14, and CK17 are hallmark features of TNBC[4]. Interestingly, cytokeratins serve more than mere epithelial cell markers, and several recent studies have provided evidence for active keratin involvement in cancer cell invasion and metastasis[20]. *Kevin* et al., in their diligent studies, showed that polyclonal breast cancer metastases arise from the collective dissemination of keratin 14-expressing tumor cell clusters[21]. Regulation of cytokeratin expression and their functional consequences, particularly in basal-like breast cancer metastasis, is not precise yet.

In this work, we show the differential role of EZH2 catalytic (elevated H3K27me3 or H3K27me3[High]) versus non-catalytic EZH2 (NC-EZH2) function in the context of TNBC progression. Interestingly, we detect that elevated H3K27me3 but not its NC-EZH2 protein overexpression results in the robust increase of TNBC peritoneal metastasis. Transcriptome analysis of H3K27me3[High] TNBC cells leads to the discovery that *KRT14* is a target of H3K27me3. Instead of H3K27me3 mediated classical transcriptional repression, we find H3K27me3 promotes *KRT14* gene expression by altering the recruitment pattern of its transcription factor SP1. Our sequential in vivo imaging experiments clearly demonstrate that KRT14 is a critical regulator of TNBC splenic metastasis, and loss of KRT14 even in H3K27me3[High] background compromises TNBC peritoneal metastasis. Further, we explore the clinical relevance of our finding in the Breast Cancer (Yau 2010) database where ER, PR, and HER2 negative basal-like breast cancer or TNBC selectively displays a positive correlation between EZH2 and KRT14 expression and H3K27me3 and KRT14 expression are found to be significantly upregulated in human TNBC metastasis as compared to their respective primary tumors. Finally, EZH2 functional loss, either by knockdown or the treatment of H3K27me3 selective inhibitor EPZ6438, significantly reduces TNBC migration, invasion in vitro, and peritoneal metastasis in vivo. In summary, our data indicate that the trimethylation function of EZH2 may drive peritoneal metastasis of TNBC, and targeting the methyltransferase activity of EZH2 could be a good option to prevent TNBC metastasis.

## Results

### EZH2 functional activation but not its protein overexpression promotes TNBC metastasis

Our studies and recent literature suggest that EZH2 plays a critical role in TNBC pathophysiology[19,22,23]. TNBC is a highly heterogeneous group of cancers, where intrinsic subtype analysis suggests approximately 80% of TNBC belongs to the basal-like category[4,24]. The basal-like or basaloid TNBC (B-TNBC) is one of the most aggressive, therapy-resistant, and metastatic tumors[25]. Therefore, we focus our entire studies on the basal-like TNBC subtype. Here, we sought to dissect the role of NC-EZH2 versus H3K27me3 in TNBC progression. We have established a 4T-1 based animal model that closely mimics basal-like TNBC progression under a preclinical setting[26]. To explore the individual role of NC-EZH2 versus H3K27me3 function, we made 4T-1 stable cells either overexpressing EZH2 catalytically inactive (−ΔSET) or catalytically hyperactive EZH2 (Y641-F) protein (Figs. 1A and 1B). Following subcutaneous and orthotopic inoculation of control and genetically modified 4T-1 cells into mice, we observe that Y641-F tumor-bearing mice have significantly smaller tumors compared to control tumor-bearing mice, while EZH2 (−ΔSET) tumor-bearing mice have no significant change in the tumor growth compared to control tumor-bearing group (Fig. 1C, D and Supplementary Fig. 1) however, Y641-F tumor-bearing mice lost marked body weight compared to control, during tumor progression (Fig. 1E). Significant body weight loss in Y641-F tumor-

bearing mice prompted us to check the metastasis status of the particular group. To understand the metastatic potential of individual groups of cells, we adapted the same system, but here we used Luc-tagged 4T-1 cells and investigated their metastatic progression through sequential live animal imaging. As shown in Fig. 1F (top and bottom panels) and 1G, Y641-F tumor-bearing mice demonstrate a substantial increase in metastasis compared to the control. Therefore, accelerated metastasis may be the reason for the marked weight loss of Y641-F tumor-bearing mice. Altogether, in-vivo studies established that H3K27me3 but not NC-EZH2 protein regulates TNBC metastasis. Increased metastasis in Y641-F cells encouraged us to explore further the impact of H3K27me3 on TNBC cell migration and invasion. Here, we performed wound healing and trans-well chamber assay to assess migration and invasion, respectively. In the wound healing assay (Fig. 1H, I), we detect Y641-F mutant cells have clear higher migratory potential as these cells are found to close the wound more rapidly than wild-type cells. Similarly, in trans-well chamber assay, Y641-F cells also show higher invasive capabilities than that wild-type cells (Fig. 1J, K). To further confirm H3K27me3-selective induction of TNBC migration, we made UTX or KDM6A (H3K27me3 specific demethylase[27]) knock down stable cells that display a selective increase of H3K27me3 without altering EZH2 levels as compared to control cells (Fig. 1L). When we assessed the migration capabilities of control and UTX KD cells by invasion assay, we observed that UTX KD cells migrate faster than control cells, again confirming H3K27me3 dependent induction of TNBC migration (Fig. 1M, N). To further rule out the contribution of EZH2 protein in modulating TNBC migration, we took two different strategies. Several studies have shown that the EZH1-SET domain has weak enzymatic activity than the EZH2-SET domain[28]. We next examined if the domain-swap approach could be useful to analyze the findings linked with NC EZH2[29]. In our first strategy, we made stable TNBC cells having either wild-type EZH2 overexpression or EZH2 overexpression containing the SET domain of EZH1 (hereafter referred to as EZH2[EZH1-SET]) and evaluated their level of EZH2, H3K27me3, and their respective migration capabilities. As shown in Fig. 1O, cells bearing EZH2[EZH1-SET] construct show overexpression of EZH2 at the protein level but H3K27me3 remains unchanged as compared to control and exhibits no change in invasion capabilities of TNBC whereas WT EZH2 overexpression results in increased invasion (Fig. 1P, Q) suggesting NC EZH2 has no role in TNBC migration induction. In our second strategy, we made use of the FDA-approved drug tazemetostat (EPZ6438), a pharmacological inhibitor of EZH2 which selectively inhibits H3K27me3 activity of EZH2 without altering its expression at the protein level (Fig. 1R). Before assessing its impact on TNBC migration, we checked its cytotoxic impact on TNBC cells and observed that it does not pose any cytotoxic impact on TNBC cells, tested up to 40 µM concentration as determined by MTS and colony formation assay (Supplementary Fig. 2). Next, we sought to determine the effect of EPZ6438 on TNBC invasion and observed that the selective inhibition of H3K27me3 by EPZ6438 at even a 10 µM dose robustly inhibited TNBC invasion compared to control (Fig. 1S, T). Altogether, our extensive in vitro and in vivo experiments clearly suggest that H3K27me3 promotes TNBC migration, invasion, and metastasis.

### Catalytically hyperactive EZH2 (elevated H3K27me3) alters the metastatic landscape of TNBC

To further validate the correlation between hyperactivation of H3K27me3 with rapid TNBC metastasis, we adapted three different strategies as described in (Fig. 2A). Interestingly, in our uniquely designed three (mixed single flank mammary fat pad, mixed tail vein, and individual double flank mammary fat pad) strategies, we observe that TNBC metastasis is robustly augmented upon H3K27me3 elevation (Figs. 2B and 2C) as compared to control. Next, we sought to

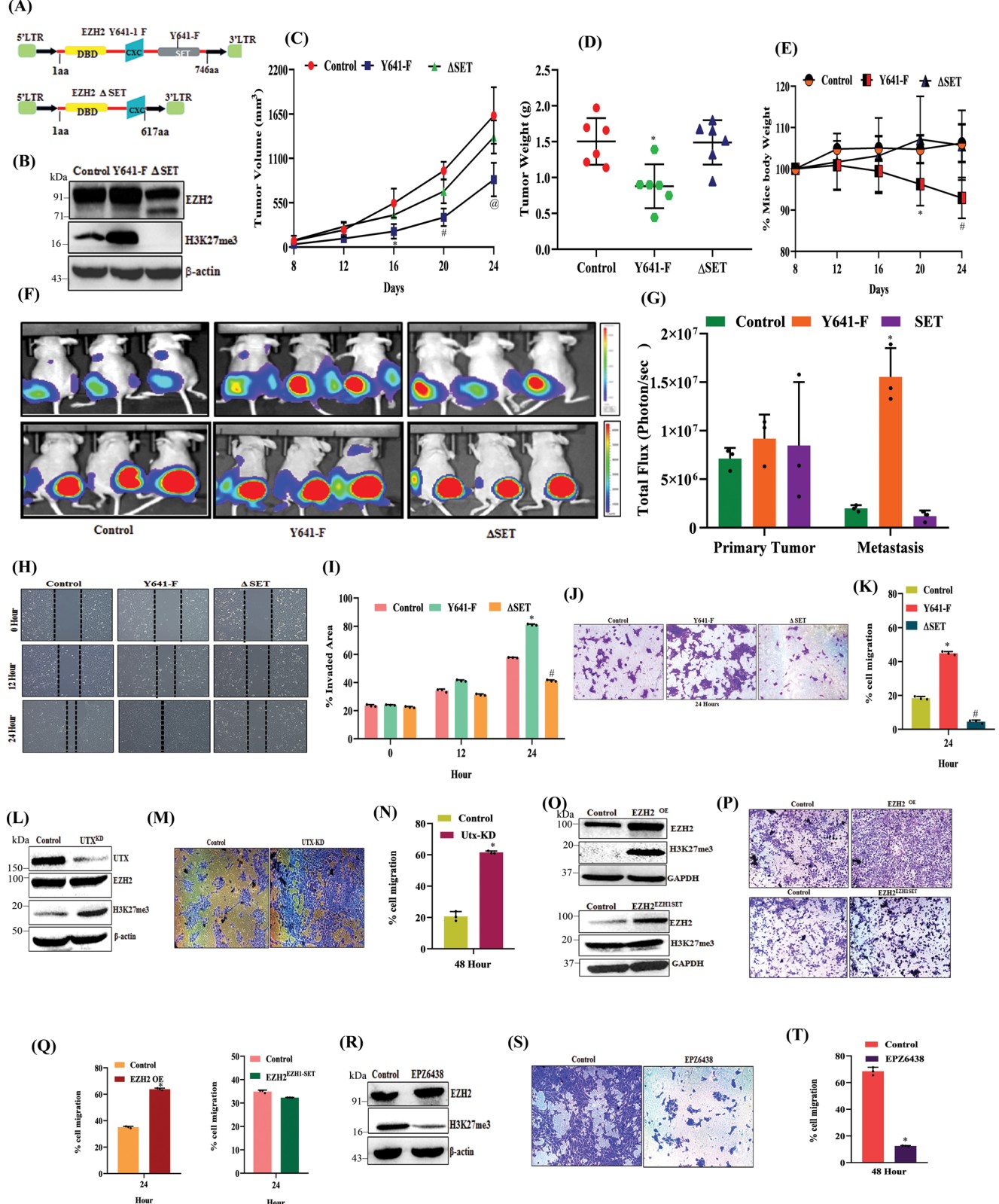

determine the TNBC metastatic pattern under the influence of increased H3K27me3. Fluorescence imaging analysis of harvested organs from double flank mammary fat pad inoculated (Control versus Y641-F) mice exhibit that control cells tend to metastasize mainly into the lungs, whereas, Y641-F cells tend to significantly metastasize into the spleen and liver (Fig. 2D). Representative fluorescent microscopic images and FACS analysis of single cells harvested from different

metastatic organs of experimental mice clearly demonstrate the striking alteration of the metastatic pattern observed with induction of H3K27me3 (Fig. 2E–G). To further validate our results in situ, we performed H&E staining of lung, liver, and spleen tissues harvested from control and Y641-F tumor-bearing mice and observed that the majority of metastatic foci are localized within the lung of control tumor-bearing mice, whereas Y641-F tumor-bearing mice display

**Fig. 1 | H3K27me3 but not NC-EZH2 protein promotes TNBC metastasis.**
**A** Schematic representation of EZH2 (Y641-F) and EZH2−ΔSET retroviral over-expression constructs. **B** Immunoblot analysis for EZH2, H3K27me3, and β-actin expression in control, stable EZH2 (Y641-F), and EZH2 (ΔSET) overexpressed (OE) 4T-1 cells. **C** Tumor growth curve is shown in control, EZH2 (Y641-F) OE, and EZH2 −ΔSET OE; data are represented as mean, error bar ± SD, left to right $^{*}P = 0.0064$, $^{#}P = 0.0001$, $^{@}P = 0.0025$, respectively, compared to control, two way ANOVA, Dunnett's multiple comparisons test. **D** Average weight of harvested tumors of Control, EZH2 (Y641-F) OE, and EZH2−ΔSET OE groups depicted by the graph, ($n = 6$) each group, data are represented as mean ± SD, $^{*}P = 0.0068$, compared to control, one way ANOVA, Dunnett's multiple comparisons test. **E** The body weight curve is shown for control, EZH2 (Y641-F) OE, and EZH2 −ΔSET OE; ($n = 5$) each group, data are represented as mean, error bar ± SD, E: left to right $^{*}P = 0.0102$ and $^{#}P = 0.013$, respectively, compared to control, two way ANOVA, Dunnett's multiple comparisons test. **F** In vivo bioluminescence monitoring of orthotopic control (left panel), EZH2 (Y641-F) OE 4T-1 Luc2- GFP (middle panel) and EZH2 −ΔSET OE (right panel) primary tumor and distant metastatic sites. The color scale indicated the photon flux (photon/ sec) emitted from each group. **G** Quantitative bar graph representation of total photon flux calculated from the region of flux (ROI), compared to control, ($n = 3$) each group, data are represented as mean, error bar ± SD, $^{*}P = 0.0021$, compared to control, two way ANOVA, Turkey's multiple comparisons test. **H** Representative images of the wound healing assay to measure the migration ability of control and EZH2 (Y641-F) OE and EZH2 −ΔSET OE 4T-1 cells in time point manner (0, 12, and 24 h), magnification 10×. **I** The quantitative analysis of wound healing assay; $^{*}P = 0.0001$ and $^{*}P = 0.0002$, two-way ANOVA, Dunnett's multiple comparisons test. **J** Representative images of the trans-well chamber migration assay to measure the invasion ability of control and EZH2 (Y641-F) OE and EZH2−ΔSET OE 4T-1 cells at 24 h. **K** The quantitative bar graphs of tans-well chamber migration assay of control and EZH2 (Y641-F) OE and EZH2 −ΔSET OE 4T-1 cells in 24 h. $^{*}P = 0.0001$ and $^{#}P = 0.0001$, one-way ANOVA, Dunnett's multiple comparisons test. **L** Immunoblot analysis for UTX, EZH2, H3K27me3, and β-actin expression in control and UTX KD 4T-1 cells. **M** Representative images of the trans-well chamber migration assay to measure the invasion ability of control and UTX KD 4T-1 cells at 48 h. **N** The quantitative bar graphs of tans-well chamber migration assay of control and UTXKD 4T-1 cells at 48 h; $^{*}P = 0.0001$, Student's t-test (two-sided). **O** Immunoblot analysis for EZH2, H3K27me3, and GAPDH expression in control, EZH2 $^{OE}$, and EZH2 $^{-EZH1 SET}$ HCC1806 cells. **P** Representative images of the trans-well chamber migration assay to measure the invasion ability of control, EZH2 $^{OE}$, and EZH2 $^{-EZH1 SET}$ HCC1806 cells at 24 h. **Q** The quantitative bar graphs of tans-well chamber migration assay of control and EZH2 $^{OE}$ and EZH2 $^{-EZH1 SET}$ cells at 24 h. $^{*}P = 0.0001$, Student's t-test (two-sided). **R** The 4T-1 cells were either treated for a vehicle or 10 μM EPZ6438 to analyze the expression of EZH2, H3K27me3, and β-actin by Immunoblot. **S** Representative images of trans-well chamber assay to analyze the invasion ability of control and EPZ6438 (10 μM) treated 4T-1 cells. **T** The quantitative bar graphs of tans-well chamber migration assay of control and EPZ6438 (10 μM) at 48 h. $^{*}P = 0.0006$, Student's t-test (one-sided). In **I**, **K**, **N**, **Q**, and **T**, Columns are the mean of triplicate readings; error bar ± SD. P values are calculated as compared to the control. Source data are provided as a Source Data file.

---

predominant liver and spleen metastasis (Fig. 2H). Further, we collected peritoneal fluid of both control and Y641-F tumor-bearing mice and cultured it for 48 h in the presence of 6TG and observed 4T-1 colonies only in the case of peritoneal fluid isolated from Y641-F tumor-bearing mice (Fig. 2I). Collectively, these series of in vivo studies strongly suggest that the enhanced global trimethylation has a significant impact on TNBC peritoneal metastasis.

## Transcriptome analysis identified *KRT14* as a target of H3K27me3 in TNBC

Next, we sought to find the correlation between H3K27me3 function and TNBC metastasis at the basal state. We orthotopically inoculated WT 4T-1-Luc cells in the mammary fat pad of female nude mice (Fig. 3A). We isolated metastatic cells from different organs following mammary fat pad inoculation and after EZH2 expression analysis; we observed that splenic metastatic cells have the highest expression of global H3K27me3 compared to cells isolated from primary tumors, followed by liver and lung (Fig. 3B). Notably, the splenic metastatic cells matched the phenotype with Y641-F cells at the basal state. Further, we inoculated splenic isolated metastatic and primary tumor cells into the mammary fat pad again and evaluated the TNBC progression. As shown in Fig. 3C (left panel), live animal imaging data clearly establishes that splenic metastatic cell inoculation results in a visible increase of TNBC peritoneal metastasis compared to the respective control (Fig. 3C, right panel). Further, splenic metastatic cell tumor-bearing mice die significantly earlier than the control (Fig. 3D).

H3K27me3-driven robust changes in TNBC metastatic signature encourage us to identify the mechanistic insight behind the phenotype. We perform differential Transcriptome analysis in control and Y641-F cells to explore this idea. In our transcriptome analysis, a total of 21044 common genes were mapped and found to be differentially expressed in control and Y641-F cells. Volcano plot analysis of the differentially expressed genes (DEGs) with ($\log_2FC$ +1, −1, and $p > 0.05$) criteria revealed 2142 significant DEGs in the Y641-F group compared to the control. Out of 2142 DEGs, 805 genes were significantly upregulated, and 1337 genes were significantly downregulated (Fig. 3E). Gene set enrichment analysis (GSEA) was performed to evaluate the characteristic dynamic pattern between control and Y641-F cells. In the gene marker section of the GSEA, the heat map was generated for the top 50 features of each phenotype (control vs Y641-F) (Fig. 3F). The heat map displayed 100 top-ranked genes (50 top-ranked high

expressions and 50 top-ranked low expressions) differentially expressed between control and Y641-F. Correlating with our observed phenotypes in vitro and in vivo, these top 100 ranked genes were explored in the literature to understand their role in cancer cell migration and metastasis. Based on the literature, a total of 16 genes were selected according to their association with cell migration and metastasis (Supplementary Table 1). A heat map of these 16 genes was plotted to show the differential expression in the control and Y641-F phenotype (Fig. 3G). Next, we individually validated the differential expression of the selected genes by the qRT-PCR between control and Y641-F cells. Of note, the expression of *NIFK*, *ADAMTS1*, *CCN2*, *JAG1*, *SEMA3C*, *TM4SF1*, *CYP1B1*, *KRT14*, and *KRT16* were found to be upregulated (Fig. 3H, Left panel), whereas, the expression of *NCAM1*, *AQP1*, *BNIP3*, *CBS*, and *NDRG1* were downregulated in Y641-F cells as compared to control cells (Fig. 3H, right panel). Next, to understand the contribution of these changes in ascertaining peritoneal metastasis in TNBC at basal state (scheme represented in Fig. 3A), we examined the mRNA expression of the validated up and downregulated genes in the metastatic cells isolated from the primary tumor, lungs, liver, and spleen. After extensive validation, *KRT14* was found to be the markedly upregulated gene in the cells isolated from lung, liver, and splenic metastasis as compared to primary tumor cells, though the highest robustness was observed in the case of splenic metastasis (Fig. 3I, left and right panels). Our previous result suggested that the global H3K27me3 level was robustly high in the metastatic cells isolated from the spleen compared to primary tumor cells (Fig. 3B). Similarly, *KRT14* is the only robustly upregulated gene in Y641-F cells compared to control cells in vitro (Fig. 3H, left panel) and primary tumor versus splenic metastatic cells in vivo, following orthotopic inoculation of wild-type cells (Fig. 3I, left panel). However, except for *KRT14*, none of the other genes passed through this validation process.

## H3K27me3 modulates *KRT14* expression at both the mRNA and protein levels

Seminal work by Lehman et al. demonstrated the importance of the *KRT* gene family in classifying TNBC, including *KRT14*[4]. Here, we sought to validate the impact of H3K27me3 on *KRT14* expression regulation. In 4T-1 cells, following overexpression of either wild-type EZH2 or Y641-F mutant EZH2, we found that KRT14 expression is significantly upregulated in EZH2$^{OE}$ and Y641-F cells as compared to controls; however, in EZH2$^{EZH1 SET}$ cells, we found no significant change in KRT14

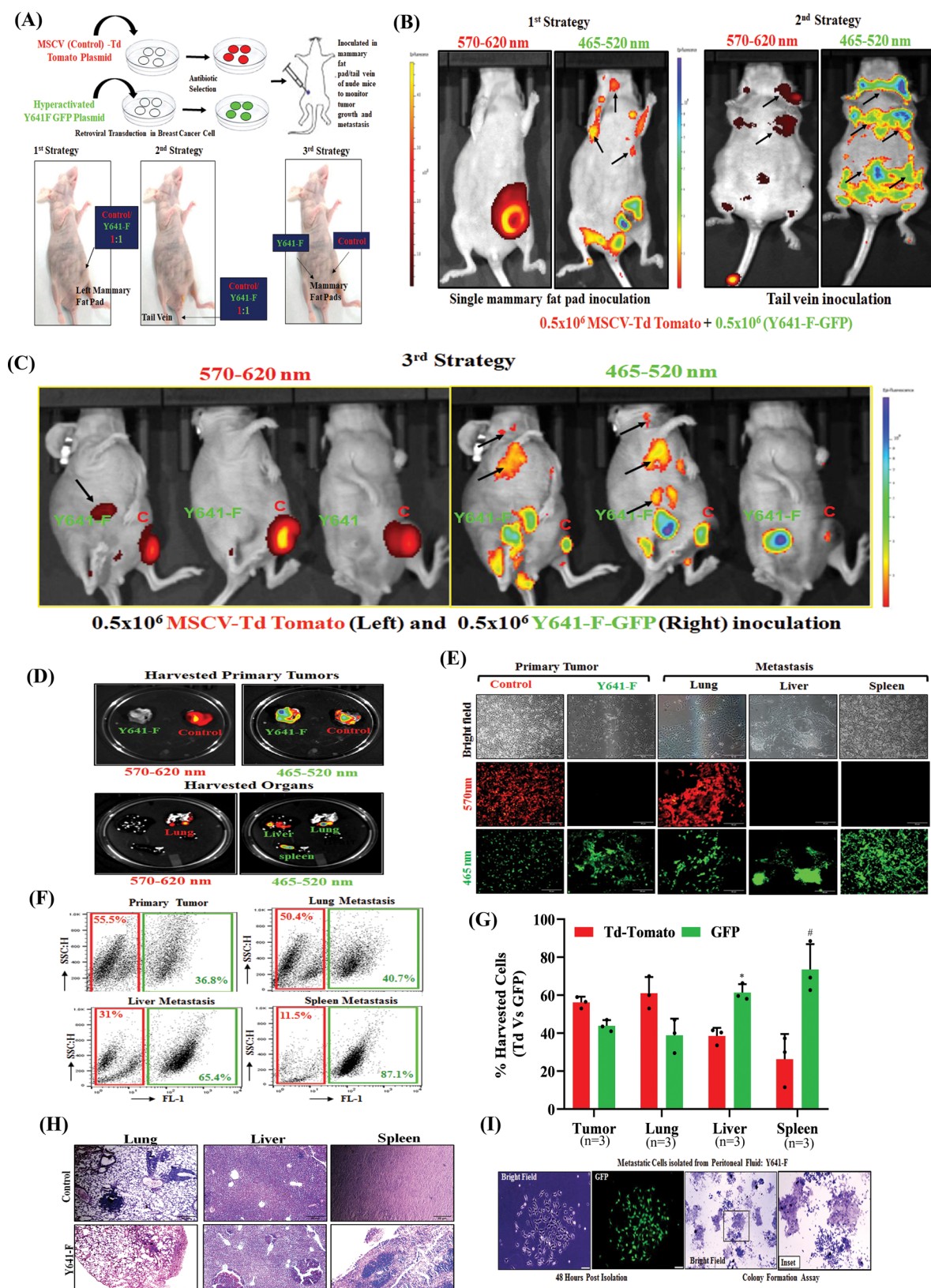

expression at the protein level (Fig. 4A–C) confirming that the H3K27me3 function of EZH2 is one of the crucial factors for *KRT14* upregulation. Moreover, Tazemetostat (EPZ6438) treatment in both control and Y641-F cells resulted in an observable reduction in KRT14 expression at both the mRNA and protein levels (Fig. 4D, E). Inducible EZH2 gain of function in HCC1806 results in an increase in KRT14 level

(Fig. 4F), while functional EZH2 inhibition by EPZ6438 in HCC1806 cells markedly down-regulates the *KRT14* mRNA expression (Fig. 4G). Further, we extend our validation experiments in two (HCC1806 and MDAMB468) different basal human TNBC cell lines. Subsequently, we found that the KRT14 protein expression was significantly down-regulated in HCC1806 and MDAMB468 by EPZ6438 treatment.

**Fig. 2 | Functional hyper-activation of EZH2 (increased H3K27me3) leads to TNBC splenic metastasis. A** Schematic representation for generation of control Td-Tomato⁺ and H3K27me3^High (Y641-F) GFP ⁺ 4T-1 cells by retroviral transduction (upper panel). The schematic delineation of different strategies used in the in vivo studies (lower panel). **B**, **C** Live fluorescence imaging of primary tumors and distant metastatic sites were shown under different strategies (described in Methods, Animal studies section). The excitation and emission wavelength: Td-Tomato- 570-620 nm and GFP-465-520 nm. **D** The fluorescence imaging of harvested control (Td-Tomato⁺) and EZH2 Y641-F (GFP⁺) primary tumors at a specific wavelength (upper panel). The metastatic potential of control (Td-Tomato⁺) and EZH2 Y641-F (GFP⁺) cells was analyzed by fluorescence imaging in the harvested organs at a specific wavelength (lower panel). **E** The fluorescence images of single-cell harvested from both tumors control (Td-Tomato⁺) and EZH2 Y641-F (GFP⁺) and harvested organs (*n* = 3). Scale bar 50 µm. **F** Representative flow cytometry-derived scatter plots showing control (Td-Tomato⁺) and EZH2 Y641-F (GFP⁺) cells in primary tumors and metastatic organs harvested from mice (*n* = 3). **G** The quantitative analysis of the percentage of Control (Td-Tomato⁺) and EZH2 Y641-F (GFP⁺) cells in harvested primary tumors and metastatic organs (*n* = 3), data are represented as mean, error bar ± SD, *P = 0.0120 and #P = 0.00484, respectively, compared to Td-Tomato group, two way ANOVA, Sidak's multiple comparisons test. **H** Photomicrographs of H&E staining in the lung, liver, and spleen of control and Y641-F tumor-bearing mice. Scale bar 50 µm. **I** Photomicrographs of metastatic cells isolated from the peritoneal fluid of Y641-F tumor-bearing mice under bright field (extreme left) and GFP fluorescence (middle left) panels, respectively. Scale bar 50 µm. The excitation and emission wavelength: GFP-465-520 nm. Colony formation assay was carried out in metastatic cells isolated from the peritoneal fluid of Y641-F tumor-bearing mice, stained with crystal violet, and photo-micrographic images were taken under bright field (middle right) and inset (extreme right) panels. Source data are provided as a Source Data file.

(Fig. 4H). The down-regulation of KRT14 was also observed in the EPZ6438 treated HCC1806 cells by confocal microscopy (Fig. 3I). After extensive in vitro validation, we sought to determine the correlation between H3K27me3 and KRT14 expression in vivo by confocal microscopy in isolated primary tumor cells versus splenic metastatic cells. As shown in Fig. 4J, the expression of both H3K27me3 and KRT14 increases in splenic metastatic cells compared to primary tumor cells, confirming the positive correlation between H3K27me3 and KRT14 in TNBC splenic metastasis in a preclinical animal model. However, we did not find any significant alteration in the expression of the proliferation marker (Ki67) between primary tumor cells and splenic metastatic cells, suggesting that an increase in H3K27me3 level only affects TNBC migration but not proliferation (Fig. 4K). These results strongly put forward that EZH2 methyltransferase activity enhances KRT14 expression in TNBC cells in vitro and in vivo.

## H3K27me3 enrichment in the *KRT14* promoter is associated with its active transcription

Classically, the H3K27me3 mark acts as a transcriptional repressor[30]. However, our data undeniably indicate that it positively regulates KRT14 expression, particularly in the basal-like TNBC subtype. To investigate whether H3K27me3 marks directly occupy the *KRT14* promoter that leads to transcription activation, we perform the chromatin immunoprecipitation (ChIP) assay in 4T-1 and HCC1806 cells by using H3K27me3 and P-S5- RNA polymerase II specific antibody. To delineate the enrichment of H3K27me3 marks in the *KRT14* promoter, we design the walking primers in the sliding window of −3000 bases upstream and +2500 bases downstream of the transcription start site (TSS) by utilizing the Eukaryotic Promoter Database based on the *KRT14* chromosomal location (Fig. 5A). The q-PCR analysis in the walking primer experiments reveals the strong enrichment of H3K27me3 marks upstream of −0.2 kb and −0.5 kb region in mouse 4T-1 cells, while in human HCC1806 cells, the enrichment of H3K27me3 marks is selectively identified only in −0.2 kb region (Fig. 5B, C). To determine whether H3K27me3 enrichment in the *KRT14* promoter is associated with transcription activation or repression, we perform the ChIP q-PCR for the transcription initiation specific RNA Pol-II C-terminal domain Ser-5 phosphorylation. Interestingly, the recruitment of p-Pol-II-S5 on the −0.2 kb region of the *KRT14* promoter is observed, indicating the transcriptional activation in both 4T-1 and HCC1806 cells (Fig. 5B, C). To further confirm the H3K27me3-mediated transcriptional activation of *KRT14*, we compare the H3K27me3 enrichment in the control and Y641-F cells by ChIP-qPCR. Most likely, the robust H3k27me3 and p-Pol-II-S5 enrichment were found in Y641-F cells in comparison to the control cells (Fig. 5D, E). Reduction of both H3K27me3 marks and p-Pol-II-S5 is identified in the *KRT14* promoter following EPZ6438 (10µM) treatment in HCC1806 cells, further establishing the authenticity of the earlier observations (Fig. 5F). Though the H3K27me3 mark classically acts as a repressor, surprisingly, we find in our case, it promotes *KRT14*

gene transcription. Besides its catalytic (H3K27me3) function, the EZH2 protein has been shown to transactivate genes in multiple elegant studies[17,18]. Therefore, we explored the EZH2 recruitment in the *KRT14* gene by utilizing the EZH2 antibody to perform a ChIP assay followed by ChIP-qPCR. However, we did not find any EZH2 enrichment in the *KRT14* promoter with respect to H3K27me3 enriched sites that further confirmed the specific involvement of EZH2 catalytic product (H3K27me3) but not EZH2 protein for the positive transcriptional regulation of *KRT14* in both 4T-1 and HCC1806 cells (Supplementary Fig. 3A, B). As the H3K4me3 mark has been associated with transcriptional activation of genes[31], next, we assess its possible involvement in *KRT14* transcriptional upregulation in the presence and absence of increased H3K27me3 state by ChIP-qPCR. We observed the significant H3K4me3 enrichment selectively in the +2.4 kb region from TSS in the *KRT14* gene in both 4T-1 and HCC1806 cells, suggesting the transcriptional progression of *KRT14* (Fig. 5G, H). Similarly, we found the marked H3K4me3 enrichment in the Y641-F cells in comparison to control 4T-1 cells in the same +2.4 kb region (Fig. 5I). Further, we also find a loss of H3K4me3 mark in the +2.4 kb region of *KRT14* promoter due to the inhibition of H3K27me3 by EPZ6438 (10 µM) in HCC1806 cells (Fig. 5J). Altogether, these results suggest that though the H3K27me3 mark classically acts as a repressor; however, we observe that it promotes *KRT14* gene transcription and the H3K4me3 enrichment in the *KRT14* promoter further confirms its transcriptional activation.

## H3K27me3 enhances *KRT14* transcription by attenuating the binding of transcriptional repressor SP1 to its promoter

To gain further insight into H3K27me3 mediated *KRT14* transcriptional activation, we search the transcription factors that regulate *KRT14* gene expression through TRRUST online software https://www.grnpedia.org/trrust/[32]. We found that SP1 acts as a major transcriptional regulator of *KRT14*. The role of SP1 has been highly characterized both as a transcriptional activator and repressor[33,34]. We analyze the presence of the putative binding motif of SP1 by employing the publicly available transcription factor binding prediction software JASPAR (http://www.jaspar.genereg.net). The SP1 usually binds to the GC- box of the promoter of the genes and regulates the transcription process. By utilizing the publicly available software Eukaryotic Promoter Database (epd.vital-it.ch,) we search the GC-box in the promoter of KRT14 to confirm the sequence of the DNA binding motif of SP1 (Figs. 6A and 6B). We have identified the GC-box −116 base upstream of TSS in the *KRT14* promoter. To confirm the SP1 involvement in regulating *KRT14* expression at the transcriptional level, we checked the *KRT14* mRNA expression in both HCC1806 and 4T-1 (Y641-F) control and SP1 knockdown (KD) cells by performing qRT-PCR. The mRNA results indicate that the SP1 KD enhances the *KRT14* mRNA expression as compared to the control HCC1806 and 4T-1 (Y641-F) cells, respectively (Fig. 6C, D). We next examined the expression of SP1 in

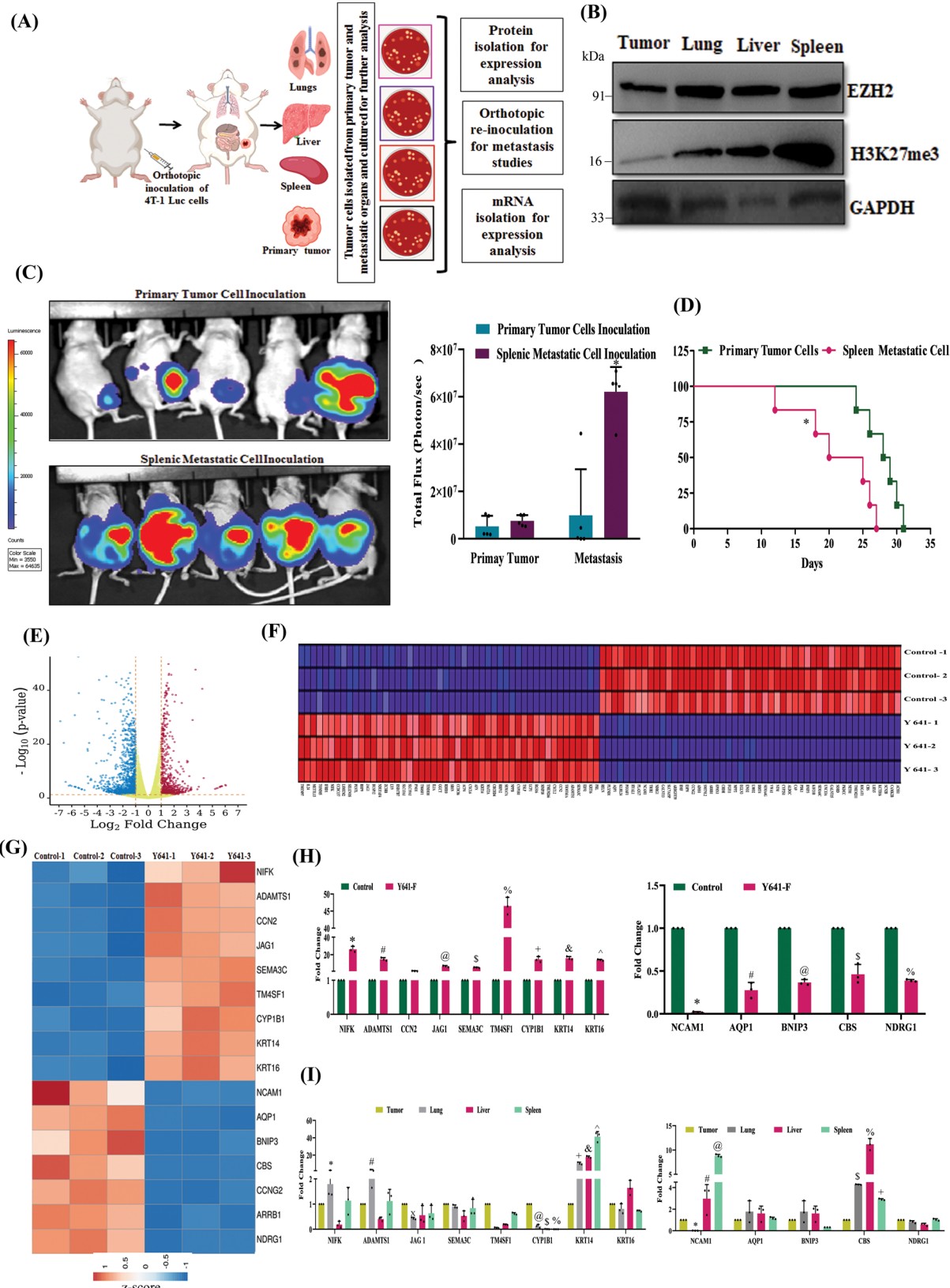

control and Y641-F 4T-1 cells via western blot and observed that the expression of SP1 remained unchanged in the Y641-F cells compared to the control (Fig. 6E). Furthermore, the western blot data also confirmed the increased KRT14 protein expression in the SP1 knockdown cells compared to control 4T-1 (Y641-F) cells (Fig. 6F). After confirming SP1 knockdown in HCC1806 cells (Fig. 6G, left

panel), we transfected *KRT14* promoter luciferase construct (−1.1 kb-consisting both proximal and distal region, and −0.5 kb consisting only proximal region) plasmids in control and SP1 KD HCC1806 cells to confirm the SP1 mediated *KRT14* transcriptional repression. As shown in Fig. 6G (right panel), a marked increase in the luciferase activity was observed in both reporter constructs in the SP1 KD cells

**Fig. 3 | Transcriptome analysis revealed that overexpression of EZH2 catalytic function (elevated H3K27me3) increases *KRT14* transcription. A** The schematic representation of experimental planning. **B** The immunoblot analysis of EZH2, H3K27me3 expression in 4T-1 cells harvested from the primary tumor and metastatic organs; GAPDH was used as a loading control. **C** The in vivo imaging of mice having a primary tumor and splenic metastatic cell inoculation ($n = 5$). The color scale indicated the photon flux (photon/s) emitted from each group. **C** Quantitative bar graph representation of total photon flux calculated from the region of flux (ROI), data are represented as mean, error bar ± SD, *$P = 0.0038$, compared to the primary tumor, two-way ANOVA, Sidak's multiple comparisons tests. **D** The Kaplan-Meier survival curve of mammary tumor onset in the nude mice with inoculation of the primary tumor and spleen metastatic cells (n = 6). *$P = 0.0150$, compared to the primary tumor, log-rank test. **E** Volcano plot represents the significant differential expression of genes between control and Y641-F cells. Red and blue dots represent the upregulated and downregulated genes with (Log$_2$FC +1, −1, and $p < 0.05$), respectively. **F** Heat map of top 100 ranked genes differentially expressed between

control and Y641-F phenotype. Expression values are represented as colors, where the range of colors (red, pink, light blue, dark blue) shows the range of expression values (high, moderate, low, lowest), respectively. **G** Heat map shows differential expression of cell migration and metastasis-related genes between control and Y641-F. **H** RNA was isolated from control and EZH2 Y641-F 4T-1 cells and subjected to real time-PCR for gene expression analysis. Data points are mean of triplicate readings of samples; error bars, ±S.D, *,#$P = 0.0001$, @$P = 0.0134$, $P = 0.001$, %,+,&,*$P = 0.0001$, two way ANOVA, Sidak's multiple comparisons test. **I** RNA was isolated from 4T-1 cells isolated from the primary tumor, metastatic lung, liver, and spleen and subjected to real time-PCR for gene expression analysis. Data points are the mean of triplicate readings of samples; error bars, ±S.D, left panel, *$P = 0.0162$, #$P = 0.0101$, X$P = 0.0031$, @$P = 0.0002$, $P = 0.0065$, %$P = 0.0039$, +$P = 0.0001$, &$P = 0.001$, *$P = 0.0001$, right panel, *$P = 0.0001$, #$P = 0.0009$, @$P = 0.0001$, $P = 0.0088$, %$P = 0.0009$ and +$P = 0.0493$, compared to isolated cells from primary tumors, two way ANOVA, Dunnett's multiple comparisons test. Source data are provided as a Source Data file.

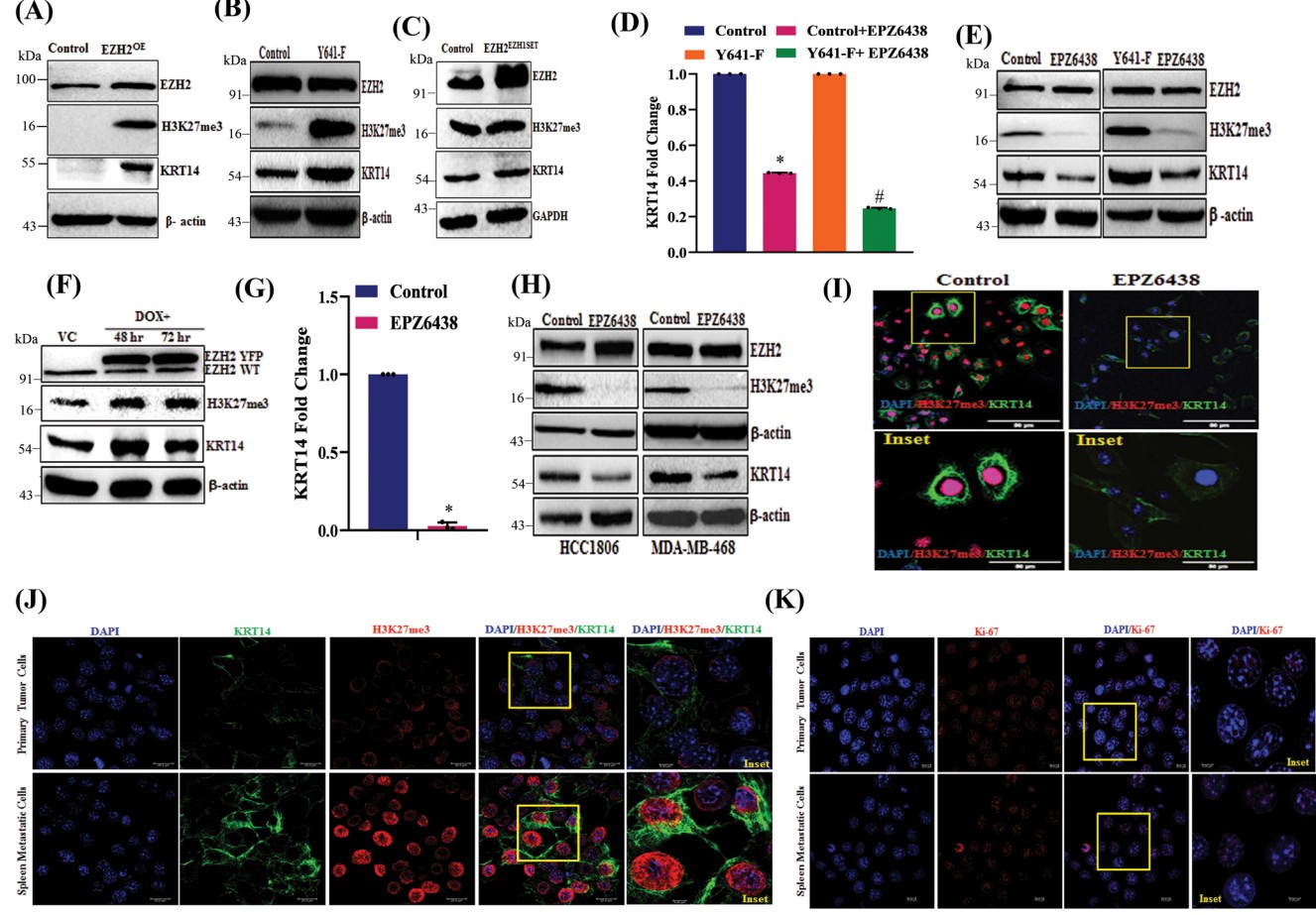

**Fig. 4 | H3K27me3 level positively regulates the expression of KRT14 in mRNA and protein levels. A** Immunoblot analysis for EZH2, H3K27me3, β-actin in control, and EZH2 $^{OE}$ HCC1806 cells. **B** Control and EZH2 Y641-F cells were analyzed for expression of EZH2, H3K27me3, KRT14, and β-actin by Immunoblot. **C** Immunoblot analysis for EZH2, H3K27me3, β-actin in control, and EZH2 $^{EZHISET}$ HCC1806 cells. **D** The control and EZH2 Y641-F cells were treated with a 10 μM dose of EPZ6438 and performed qPCR analysis for *KRT14* expression. Data points are the mean of triplicate readings of samples; error bars, ±S.D, compared to control *$P = 0.0001$ and Y641-F #$P = 0.0001$, respectively, one-way ANOVA, Turkey's multiple comparisons test. **E** The control and EZH2 Y641-F cells were treated with a 10 μM dose of EPZ6438 and analyzed for expression of EZH2, H3K27me3, KRT14, and β-actin by immunoblot. **F** The vehicle control and Tet-ON EZH2-YFP OE HCC1806 cells were analyzed for expression of EZH2 WT, EZH2-YFP, H3K27me3, KRT14, and β-actin by

immunoblot. **G** The HCC1806 cells were treated either for vehicle control or 10 μM dose of EPZ6438 for the analysis of the expression of *KRT14* by RT-qPCR. Data points are the mean of triplicate readings of samples; error bars, ±S.D, *$P = 0.0001$ compared to control cells, student's *t*-test (two-sided). **H** The HCC1806 and MDAMB468 cells were either treated for a vehicle or 10 μM EPZ6438 to check the expression of EZH2, H3K27me3, KRT14, and β-actin by immunoblot. **I** HCC-1806 cells were treated with either vehicle or 10 μM EPZ6438 treatments for 24 h, co-stained with H3K27me3 (red) and KRT14 (green) antibodies, and analyzed under a confocal microscope. Scale bar, 50 μm. **J, K** 4T-1 cells were isolated from primary tumors and metastatic spleen and subjected to confocal analysis after having either co-staining (**J**) with H3K27me3 (red) and KRT14 (green) or single (**K**) staining with Ki-67 (red) antibodies. In both cases, DAPI was used to stain the nuclei. Scale bar 20 and 5 μm. Source data are provided as a Source Data file.

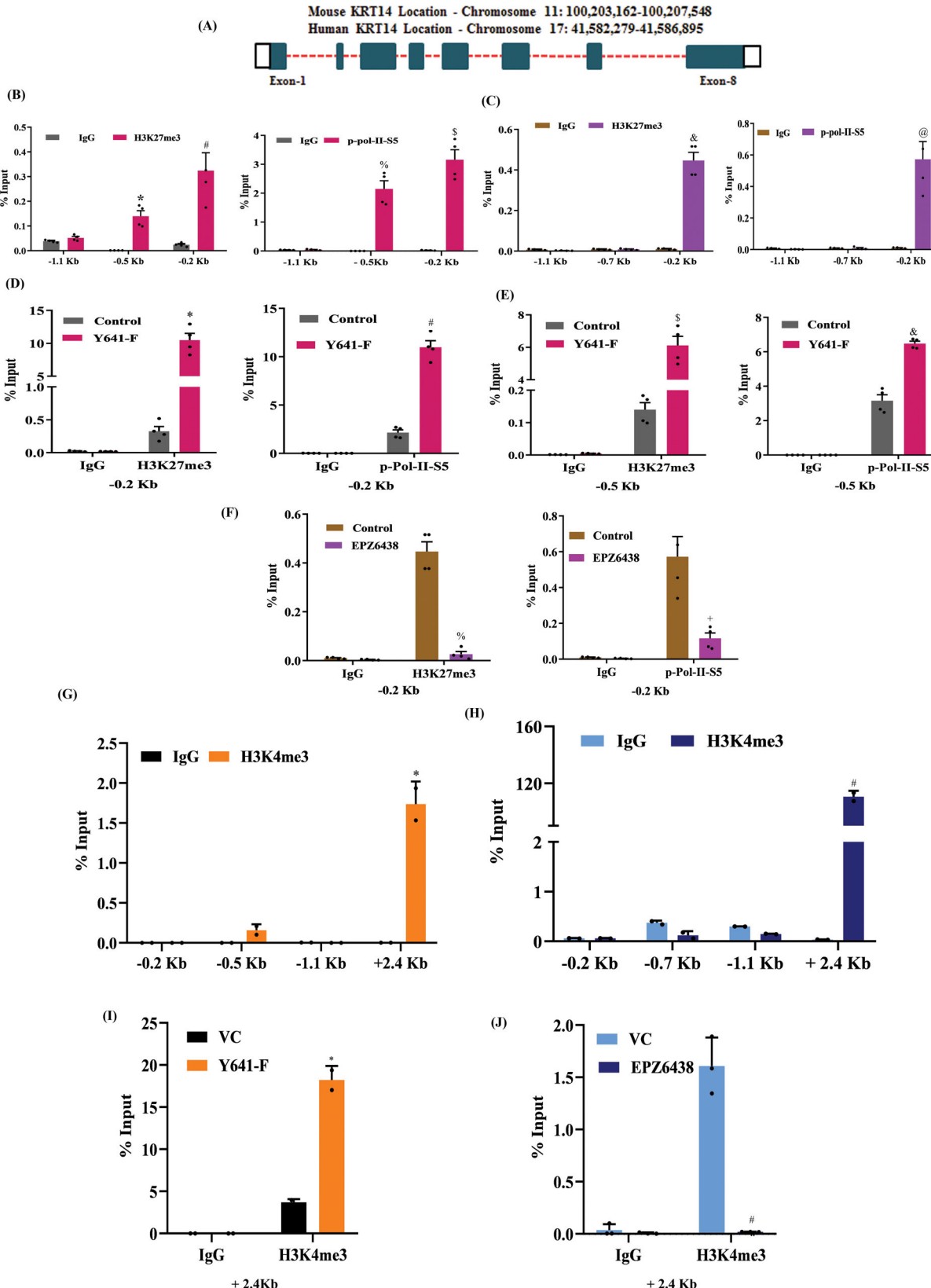

in comparison to the control suggesting −0.5 kb proximal region (consisting of GC box) is sufficient to inhibit *KRT14* transactivation. Next, we treated the HCC1806 cells either with vehicle control or two doses (150 nM and 300 nM) of SP1 inhibitor Mithramycin, (known to bind GC-rich DNA sequences and attenuate the SP1 binding) and observed mild inhibition of SP1 protein expression

(Fig. 6H, left panel). To assess the role of SP1 in *KRT14* gene trans-activation, we transfected control, and −0.5 kb *KRT14* promoter-luciferase construct plasmid in HCC1806 cells following vehicle and Mithramycin treatments. We found a noticeable increase in the luciferase activity in both doses of Mithramycin as compared to vehicle treatment (Fig. 6H, right panel). Together, these findings

**Fig. 5 | H3K27me3 occupancy in the *KRT14* promoter is linked with active transcription. A** Diagrammatic scheme showing the genomic location of the *KRT14* gene for both mice and humans. **B** ChIP was performed in 4T-1 mouse cells using anti-H3K27me3, p-Pol-II-S5, and IgG antibodies and then examined by real-time qPCR using primer pairs targeting −1.5 to −0.2 kb of the *KRT14* gene. **C** Same procedure for HCC1806 human cells as in (**B**). **D, E** The differential fold change enrichment for H3K27me3 and p-Pol-II-S5 were observed in −0.2 and −0.5 kb regions from TSS in control and EZH2-Y641-F cells by ChIP qPCR. **F** The differential fold change enrichment for H3K27me3 and p-Pol-II-S5 were analyzed in the −0.2 kb region from TSS in the HCC1806 cells treated either with vehicle control or 10 μM EPZ6438. **G** ChIP was performed in 4T-1 mouse cells using anti-H3K4me3 and IgG antibodies and then examined by real-time qPCR using primer pairs targeting −1.1 to +2.4 kb of the *KRT14* gene. **H** Same procedure for HCC1806 human cells as in (**G**). **I** The differential fold change enrichment for H3K4me3 + 2.4 kb region from TSS in control and EZH2- Y641-F 4T-1 cells by ChIP qPCR. **J** The differential fold change

enrichment for H3K4me3 was analyzed in +2.4 kb region from TSS in the HCC1806 cells treated either with vehicle control or 10 μM EPZ6438. **B–F** Columns, are the mean of quadruplicate readings of samples; error bars, ±S.D, **B, C** compared to IgG control, left to right, $^*P = 0.0158$, $^\#P = 0.0001$, $^\$P = 0.0003$, $^\$P = 0.0001$, $^\&P = 0.0001$ and $^@P = 0.0004$, two way ANOVA, Sidak's multiple comparisons test. **D–F** compared to control, left to right, $^*P = 0.0029$, $^\#P = 0.0025$, $^\$P = 0.0022$, $^\&P = 0.0024$, $^\%P = 0.0036$ and $^+P = 0.0134$, two-way ANOVA, Turkey's multiple comparisons test. **G–I** columns, a mean of duplicate readings of samples, error bars, ±S.D, **G, H** compared to IgG control, left to right, $^*P = 0.0023$ and $^\#P = 0.0001$, two-way ANOVA, Sidak's multiple comparison test. **I** compared to VC, $^*P = 0.0001$, two-way ANOVA, Turkey's multiple comparisons test. **J** Columns, a mean of triplicate readings of samples, error bars, ±S.D, compared to VC, $^\#P = 0.0082$, two-way ANOVA, Turkey's multiple comparisons test. Source data are provided as a Source Data file.

ossify that SP1 acts as a strong and direct transcriptional repressor for *KRT14*. Interestingly, the H3K27me3 and p-ser-5-Pol-II enrichment have also been recognized in the same region where SP1 binds to *KRT14*. To examine whether H3K27me3 regulates the recruitment of SP1 in the GC box, we further confirm the binding of SP1 in the GC box of the *KRT14* promoter. We performed the ChIP q-PCR in the SP1 immunoprecipitated DNA in HCC1806 control and EPZ6438 (10μM) treated cells. Here, SP1 recruitment was found to be upregulated in the EPZ6438 treated cells compared to the control (Fig. 6I, left panel), while the H3K27me3 and p-Pol-II-S5 enrichment in the *KRT14* promoter are downregulated in EPZ6438 treated cells as compared to control cells (Fig. 6I, middle and right panels). Overall, these findings suggest that the H3K27me3 may compact the GC box region in the *KRT14* promoter to inhibit the SP1 binding in the GC box and permits RNA polymerase-II to initiate the transcription of the *KRT14* gene (Fig. 6J).

## Genetic ablation of *KRT14* impairs splenic metastasis in TNBC
H3K27me3 mediated selective *KRT14* up-regulation in TNBC peritoneal metastasis encouraged us to explore the role of KRT14 in TNBC migration, invasion, and metastasis. First, we knocked down the *KRT14* in 4T-1 (Y641-F) and HCC1806 cells by using the two-specific shRNA against *KRT14* mRNA and validated their efficacy (Figs. 7A and 7B). In the wound healing assay, we find *KRT14* KD inhibits the closure of the wound as compared to the control (Fig. 7C–F). Similarly, in the transwell chamber assay, we observe *KRT14* KD cells have lower invasion capabilities than control cells (Fig. 7G–J). To determine the role of KRT14 in TNBC splenic metastasis, we perform the orthotopic single mammary fat pad mixture experiment where (Y641-F) control Td-Tomato$^+$ and (Y641-F) *KRT14* KD GFP$^+$ cells are equally mixed and inoculated in one of the mammary fat pads of mice ($n = 5$). After 25 days, we harvested the spleen, isolated the single cells from it, and allowed them to grow for three days with 6-TG selection. As shown in fluorescence microscopy pictures (Fig. 7K, L), most metastatic cells migrated towards the spleen are found to control Td-Tomato$^+$ cells compared to GFP$^+$ *KRT14* KD cells. Next, cells were trypsinized and analyzed by FACS to quantitatively analyze the percentage of Control and *KRT14* KD cells metastasized to the spleen. Our FACS analysis clearly suggests that the percentage of migrated metastatic control Td-Tomato$^+$ cells is markedly higher in the spleen as compared to metastatic Y641-F GFP$^+$ cells (Fig. 7M, N). Altogether, these studies conclusively demonstrate that loss of KRT14 expression reduces TNBC cell migration and invasion capabilities and markedly hinders TNBC splenic metastasis.

## H3K27me3 and KRT14 levels are significantly increased in human TNBC metastasis
To draw a clinical correlation between EZH2 and KRT14 expression, we mined Breast Cancer Yau (2010) dataset from breast cancer patients

and observed a high *EZH2* and *KRT14* mRNA expression exclusively in the basal (TNBC) subtype compared to the other breast cancer subtypes (Fig. 8A, B). Next, we sought to investigate the protein expression correlation between EZH2 and KRT14 in TNBC patient samples from Breast Cancer Gene-Expression Miner v4.8 and discovered the protein expression of EZH2, and KRT14 is positively ($p = 0.0018$, $r = 0.1$) correlated with each other (Fig. 8C). To address the correlation between EZH2, H3K27me3, and KRT14 expression in human TNBC distant metastasis, parallel sections of TNBC primary tumors and matched metastatic organs ($n = 10$) were stained with antibodies against EZH2, H3K27me3, and KRT14. IHC staining of TNBC primary tumors displayed moderate expression of EZH2 and KRT14, but the H3K27me3 level was found to be scanty (Fig. 8D–G). On the other hand, compared with matched primary tumors, respective metastatic counterparts display a visible increase in positive staining for H3K27me3 and KRT14 in most of the TNBC metastatic samples, whereas EZH2 positivity was found to be inconsistent (Fig. 8E–G). However, the TNBC liver metastatic sample exhibits strong positive staining for EZH2, H3K27me3, and KRT14, suggesting that EZH2 is functionally hyperactive in the metastatic liver site (Fig. 8D, lower panel).

## Inhibition of EZH2 impairs TNBC peritoneal metastasis
We wish to explore the clinical relevance of our finding further and finally understand whether EZH2 functional inhibition rescues the in vivo phenotype. In this endeavor, we confirm EZH2 knockdown in 4T-1 cells through immunoblot (Fig. 9A). Indeed, EZH2 knockdown cells display less migratory (Fig. 9B, C) and invasive (Fig. 9D, E) potential than their respective controls. Further, the serial bioluminescence imaging and analysis of harvested organs from control EZH2 KD tumor-bearing mice reveal that the depletion of EZH2 restricts the peritoneal metastasis as compared to the respective control (Fig. 9F–H). Also, compared to control, EZH2 KD increases mice survival, as shown in Fig. 9I. As EZH2 inhibitor EPZ6438 (tazemetostat) recently received FDA approval for sarcoma treatment, we readily evaluated its potential to inhibit TNBC migration, invasion, and metastasis[35,36]. As observed in Fig. 9J–M, EPZ6438 treatment results in marked inhibition of TNBC migration and invasion, compared to their respective controls. Consistent with the genetic depletion of EZH2, we observe a visible decrease of peritoneal metastasis following daily oral administration of EPZ6438 (250 mg/kg) though primary tumor burden remains the same in both the control and treatment group (Fig. 9N–P and Supplementary Fig. 4). Moreover, EPZ6438 treatment resulted in a marked increase in mice survival as compared to the vehicle-treated group (Fig. 9Q).

Overall, our in vitro, in vivo, and human patient sample data suggest that the EZH2-H3K27me3-KRT14 axis may be one of the critical regulators of TNBC peritoneal metastasis, and FDA-approved EZH2 inhibitor EPZ6438 or Tazemetostat could be a potential therapeutic option against TNBC progression (Fig. 10).

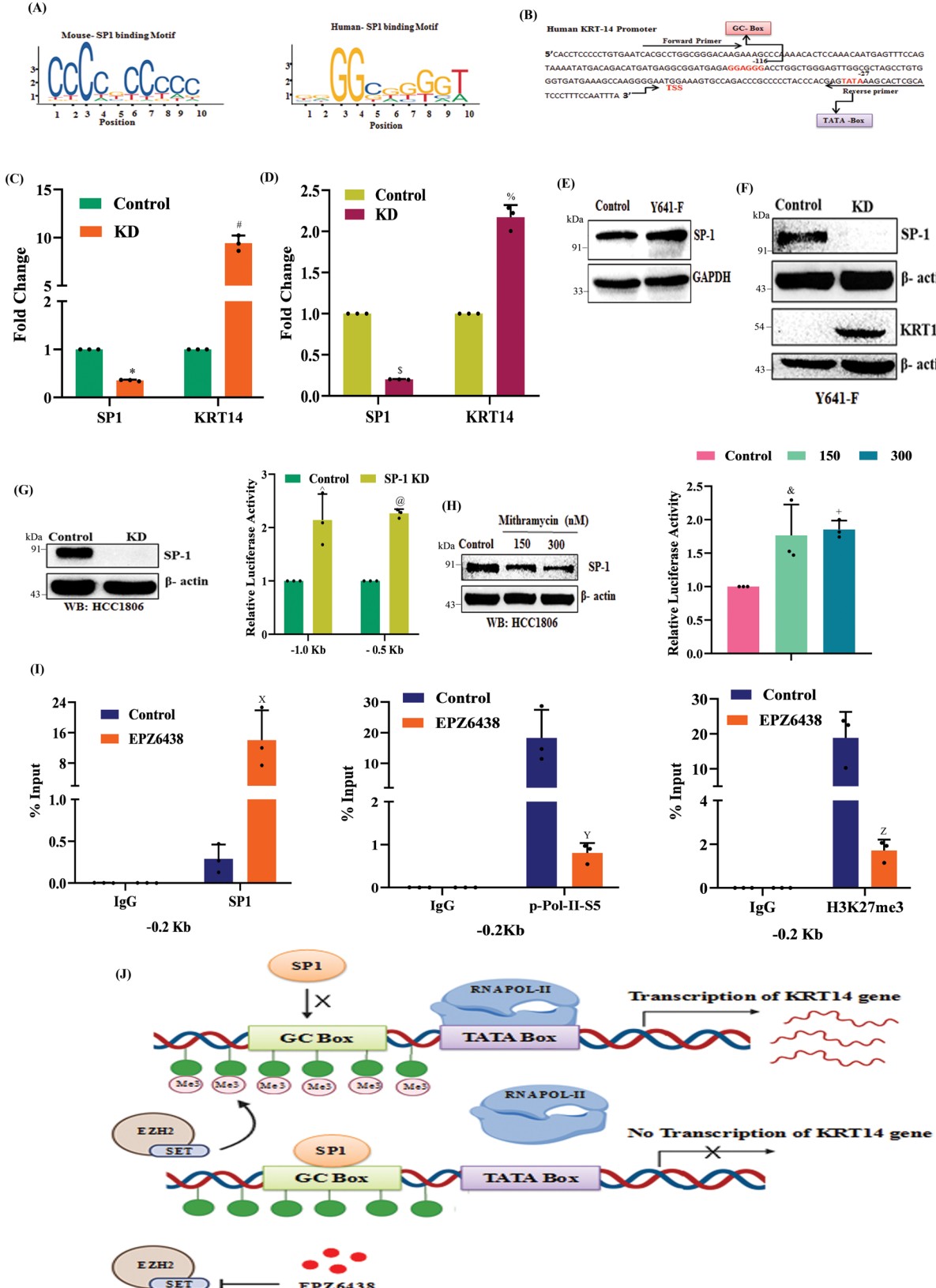

## Discussion

In the current study, we set out to dissect H3K27me3 versus NC-EZH2 in basal-like TNBC growth and progression and discover that H3K27me3 is the key to TNBC peritoneal metastasis. Transcriptome analyses and subsequent validation lead us to ascertain KRT14 as a target of H3K27me3. Importantly, we observe an increased H3K27 tri-methylation mark for transcriptional activation of *KRT14* instead of H3K27me3-mediated classical transcription repression. Loss of EZH2 or KRT14, or H3K27me3 function results in robust inhibition of TNBC peritoneal metastasis, particularly splenic metastasis. The human patient sample data also revealed a positive correlation between H3K27me3 and KRT14 expression in TNBC metastasis, including liver metastasis.

**Fig. 6 | H3K27me3 enrichment in the *KRT14* promoter attenuates SP1 binding and promotes *KRT14* expression. A** Diagrammatic representation of the SP1 binding motif obtained from the JASPAR database for a mouse (left) and human (right). **B** The human nucleotide sequence for *KRT14* promoter with TATA and GC-box. **C, D** The *KRT14* mRNA expression was analyzed in the control and SP1 KD HCC1806 (**C**) and 4T-1 (**D**) cells by qRT-PCR. left to right, compared to control, $^*P = 0.0002$, $^\#P = 0.0030$, $^\$P = 0.0001$, and $^\%P = 0.0053$, student's *t*-test (two-sided). **E** The control and Y641-F cells were analyzed for SP1 and GAPDH by immunoblot. **F** The control and SP1 KD 4T-1(Y641-F) cells were analyzed for SP1, KRT14, and β-actin expression by immunoblot. **G** Control and SP1 KD HCC1806 cells were analyzed for SP1, and β-actin protein expression by immunoblot (left panel), and the control and SP1 KD cells were analyzed for luciferase activity followed by the transfection of −1.1 and −0.5 kb *KRT14* promoters (right panel). **G** Right panel, compared to control, $^*P = 0.0010$ and $^@P = 0.0005$, two-way ANOVA, Sidak's

multiple comparisons test. **H** The HCC1806 cells were treated either with vehicle or 150 nM or 300 nM of Mithramycin and subjected to immunoblot analysis for SP1 and β-actin (left panel). Relative luciferase activity was measured as described in the Methods section and represented in the right panel. H: right panel, compared to control, $^\&P = 0.0289$ and $^+P = 0.0201$, two-way ANOVA, Dunnett's multiple comparisons test. **I** ChIP q-PCR data showing the recruitment of SP1 (left panel), p-Pol-II-S5 (middle panel), and H3K27me3 (right panel) on the *KRT14* promoter upon EPZ6438 (10 μM) treatment in HCC1806 cells. **I** Left to right, compared to control, $^XP = 0.0052$, $^YP = 0.0097$, and $^ZP = 0.0011$, two-way ANOVA, Sidak's multiple comparisons test. In, **C, D, G** (right), **H** (Right), and **I** Columns represent a mean of triplicate readings of samples, error bars, ±S.D. **J** Pictorial representation illustrating how H3K27me3 inhibits the SP1 binding to the promoter of the *KRT14* gene and activates its transcription. Source data are provided as a Source Data file.

Underpinning our observations, recently, the Mittal group performed a bioinformatic analysis of 2000 patients in the breast cancer cohort and identified that TNBC patients had shown the enhanced expression of EZH2 with an overall poor survival rate[19]. Moreover, several human tissue microarray studies correlated EZH2 with poor prognosis in TNBC[11,37,38]. In support of all the pre-existing information, our analyses further prove that, particularly in the basal-like TNBC subtype, which covers 80% of the whole group[4], EZH2 activity is a critical driver for TNBC progression. Moreover, EZH2 overexpression in breast cancer does not always correlate with increased expression of global H3K27me3[9,39,40]. It is inadequately established whether TNBC with distinct levels of EZH2 protein and its catalytic function (H3K27me3) belongs to the same or different biological behavior. Recently, the Dihua group highlighted EZH2 methytransferase-independent role in breast cancer brain metastasis. One of their studies suggests that EZH2 promotes TGFβ signaling to promote breast cancer bone metastasis via activation of integrin β1-FAK signaling[41,42] following intra-cardiac or tail vein inoculation of the breast cancer cells. However, we primarily observe peritoneal and lung metastasis but not brain metastases in our in vivo imaging experiments following orthotopic inoculation of EZH2 hyperactive breast cancer cells. These findings again suggest the highly context-dependent function of EZH2 even in regulating organ-specific metastasis of breast cancer. Peritoneal metastasis is a classical signature of TNBC metastasis in patients, which is largely missing in most of the preclinical studies as the route of tumor cell inoculation is not always orthotopic, instead, it is tail vein or intra-cardiac[43,44]. Further, it has been shown that peritoneal metastasis is associated with poor survival among the other distant metastatic sites of breast cancer[45–47]. Therefore, our observations regarding H3K27me3-dependent TNBC splenic metastasis display the actual clinical scenario, and its further inhibition may have a translational impact on TNBC pathophysiology. Previously, another compelling study showed the effect of epigenetic reprogramming in the progression of pancreatic ductal adenocarcinoma (PDAC) and identified the substantial difference in the enrichment of repressive (H3K9me3) and activating (H3K27ac and H3K36me) marks between peritoneal and distant metastasis[48]. Similarly, we also identified a massive epigenetic reprogramming in the expression of global H3K27me3 between the isolated cells from primary tumors versus cells from peritoneal metastatic organs like the spleen and liver. Further, we observe that isolated H3K27me3 enriched spleen metastatic cells have even aggressive disease progression and fatality in mice indicating the critical correlation between elevated H3K27me3 function and TNBC progression and its overall poor survival.

Our transcriptome analysis of H3K27me3^High cells not only identifies cytokeratin 14 (*KRT14*) as a fresh target of H3K27me3 in TNBC but also establishes a unique correlation between the induction of H3K27me3 with upregulation of gene (*KRT14*) expression. In support of our positive regulatory circuit between H3K27me3-KRT14, Granit et al., earlier demonstrated silencing of EZH2 protein reduces the

expression of KRT5 and KRT14, whereas promotes the expression of the luminal cytokeratin KRT18 in the course of maintaining bi-lineage identity in basal-like breast cancer[49]. Usually, EZH2-mediated silencing of genes is the fundamental mechanism of the PRC2 complex, as described by several elegant studies[17,18,30,50]. Despite its classical suppressive function, the Majewski group first documented enrichment of H3K27me3 in the transcriptionally activated genes[51]. In correlation with our finding, H3K36me3 and H3K4me2 also have a contrasting impact on gene expression depending on their differential distribution in chromatin landscape[52,53]. The evolution in the ChIP-Seq pipeline revealed that the chromatin landscape is highly dynamic, and the distribution of epigenetic marks is not universal. Multiple ChIP-seq studies advocate that the PRC2 binding is identified in a relatively very minor part of the genome; however, its catalytic product is found in 70–80% of all H3[54–57]. In support of such observations, we also did not find any enrichment of particular genes when we performed ChIP with the EZH2 antibody (Supplementary Fig. 3A, B), again suggesting that the epigenetic modulators and their biological products may not always be correlative in terms of recruitment and gene regulation. Similarly, in clinical samples, we observe a good degree of correlation between the expression H3K27me3 and KRT14 in TNBC metastasis, where EZH2 levels failed to corroborate with high H3K27me3 levels. Indeed, any epigenetic signature changes in the classical distribution pattern may change the global network of gene transcription[51–53,58]. Further extending the concept, here we provide an insight that recruitment of transcription factor can be controlled by epigenetic modulators as H3K27me3 promotes *KRT14* transcription by inhibiting the binding of transcription factor SP1 to its promoter in TNBC cells. Another essential aspect of our study is the positioning of KRT14 as one of the potential metastatic regulators in TNBC. So far, it is well known as an intrinsic molecular marker for basal-like TNBC[4,24], but our studies demonstrate its functional significance in TNBC pathophysiology. In support of our observations, several reports have suggested that the KRT14^+ cells have a metastatic advantage compared to KRT14^− cells in the TNBC heterogeneous population, and they regulate the function of a wide plethora of genes involved in different stages of metastatic cascade[21,59,60]. Further, KRT14-expressing cells act as stem cells for natural or injury-induced bladder regeneration and typically lead to the origin of bladder cancer[61]. Though the H3K27me3-KRT14 axis played a major role in delineating TNBC peritoneal metastasis in our experimental setup, the contribution of other regulatory factors, such as EZH2-mediated suppression versus activation of gene expression, should also be carefully considered in the bigger picture of TNBC metastasis.

Existing literature indicates that the role of EZH2 in tumor growth and progression is highly context-dependent, as mentioned earlier, and to some extent, it exerts an opposite effect in different types of cancers. Our recent studies in colon cancer, as well as another thorough study by Gonzalez et al., in breast cancer, reiterate that loss of EZH2 results in primary tumor growth inhibition, whereas our current

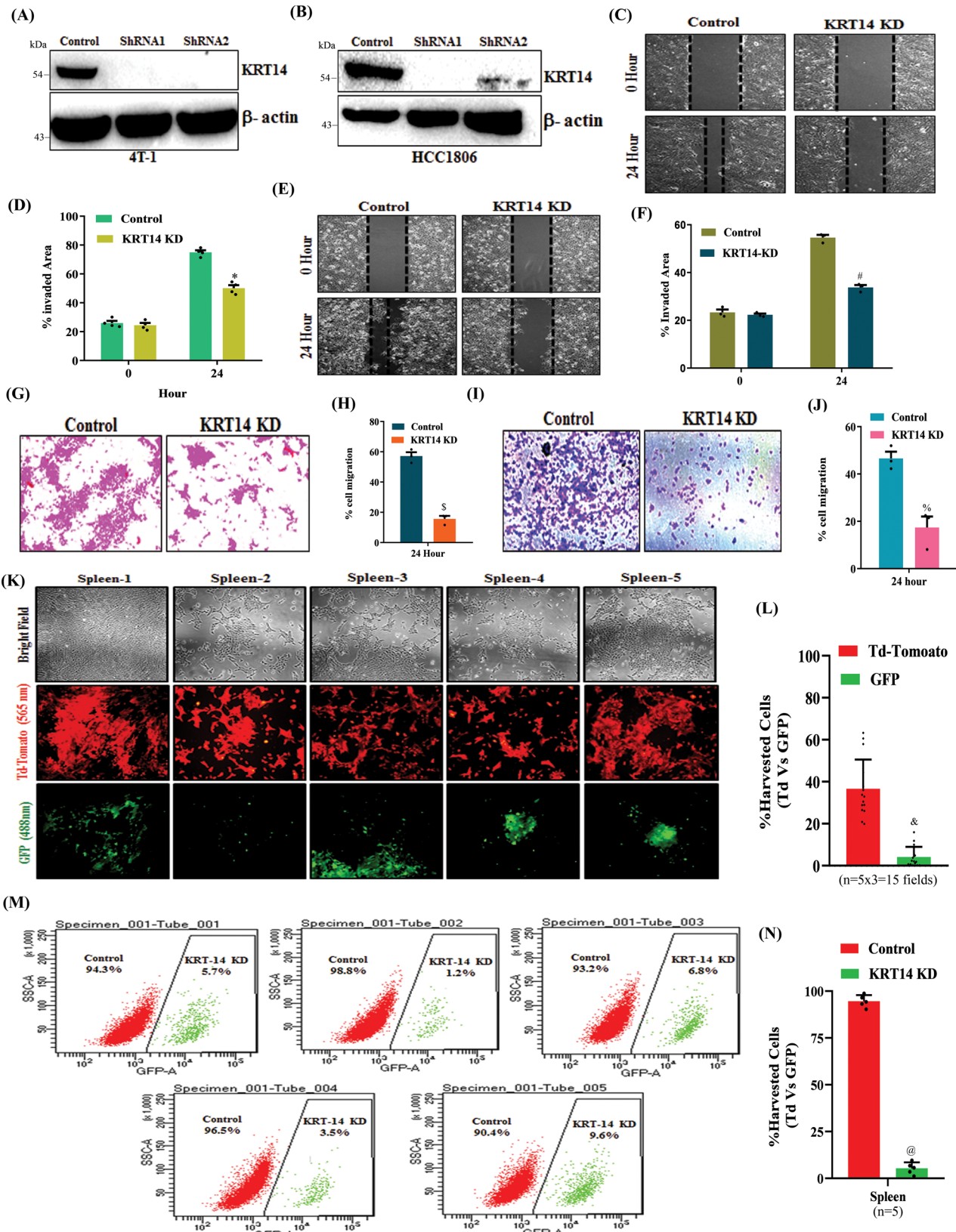

observation in TNBC shows no significant impact of EZH2 in primary tumor growth[12,62]. Consistent with our anti-metastatic effect of EZH2 inhibitor, recent studies[19,39] that tested the effect of EZH2 HMT inhibitor (GSK-126) against TNBC and luminal B breast cancer, respectively, also witnessed the robust anti-metastatic potential of EZH2 inhibitor. Targeting EZH2 with methyltransferase inhibitors has not

always been advantageous in clinical trials, apparently due to the EZH2 methyltransferase-independent functions in tumorigenesis[63,64]. Moreover, various preclinical and clinical studies suggest that the EZH2 inhibitor (Tazemetostat) showed better efficacy and minimized therapy relapse primarily in lymphomas but not in solid tumors[64]. Tazemetostat selectively inhibits the H3K27me3 function of EZH2 but not

**Fig. 7 | Genetic knockdown of *KRT14* inhibits splenic metastasis. A, B** The *KRT14* knockdown was confirmed through immunoblot by using two different shRNA in 4T-1 and HCC1806 cells. The β-actin is used as a loading control.
**C–F** Representative images of the wound healing assay to measure the migration ability of control and *KRT14* KD in 4T-1 (**C**) and HCC1806 (**E**), magnification 10 ×.
**D, F** The quantitative analysis of wound healing assay. Columns, a mean of quadruplicate readings of samples; error bar, ±SD, *P = 0.0001 and #P = 0.0001, compared to control, two-way ANOVA, Sidak's multiple comparisons test.
**G–J** Representative images of the trans-well chamber migration assay to measure the invasion ability of control and *KRT14* KD 4T-1 (**G**) and HCC1806 (**I**) cells at 24 h. The quantitative bar graphs of the trans-well chamber migration assay of control and *KRT14* KD 4T-1 (**H**) and HCC1806 (**J**) cells are shown. Columns represent a mean of triplicate readings of samples, error bars, ±S.D. compared to control, $P = 0.0002, %P = 0.0058, Student's *t*-test (two-sided). **K** Fluorescence imaging of

metastatic cells harvested from spleen (*n* = 5). The Td-Tomato fluorescence is for control cells, and GFP fluorescence is for *KRT14* KD (left panel). Scale bar 50 μm. The excitation and emission wavelength: Td-Tomato-570–620 nm and GFP-465–520 nm. **L** The quantitative analysis for the metastatic control (Td-Tomato⁺) and *KRT14* KD (GFP⁺) cells harvested from the spleen (*n* = 5, three different fields from each spleen), Columns, a mean of fifteen readings of samples, error bar, ±SD, &P = 0.0001, compared to Td-Tomato, Student's *t*-test (two-sided).
**M** Representative flow cytometry-derived scatter plots showing control (Td-Tomato⁺) and EZH2 Y641-F (GFP⁺) metastatic cells harvested from spleen (*n* = 5).
**N** The quantitative analysis of the percentage of Control (Td-Tomato⁺) and *KRT14* KD (GFP⁺) metastatic cells harvested from spleen (*n* = 5) Columns, a mean of five readings of samples, error bar, ±SD, @P = 0.0001, compared to Td-Tomato, Student's *t*-test (two-sided). Source data are provided as a Source Data file.

the level of total EZH2 protein, and lymphomas have survival dependency on H3K27me3 function[65]. The overall discrepancy in the efficacy of EZH2 inhibitors in lymphoma versus solid tumors, including TNBC in the context of primary tumor growth, is due to the lack of particular mutations (Y646, A682, and A692) in solid tumors that are present in lymphomas. These gain of function mutations result in catalytically hyperactive EZH2 in 25–27% of germinal center follicular lymphomas and make them vulnerable to EZH2 inhibition therapy. Therefore, Tazemetostat has proven to be very effective in reducing primary tumor growth in lymphomas, whereas, selective loss of H3K27me3 function by Tazemetostat does not impact primary tumor growth or tumor cell proliferation in most solid tumors. However, as observed by us and others, selective loss of H3K27me3 function of EZH2 poses a significant inhibitory influence on cancer cell migration and tumor metastasis[19]. Context dependency is profound, even in terms of EZH2-dependent gene regulation. Our breast cancer (Yau 2010) data mining in the PAM50 subtypes of breast cancer cohort display a positive correlation between EZH2 and KRT14 expression selectively in basal breast cancer subtype (TNBC) among all breast cancer subtypes. In compliance with preclinical observations, the EZH2-H3K27me3-KRT14 axis is highly overexpressed in peritoneal (liver) metastasis of human TNBC samples compared to matched primary tumors. However, in the case of other secondary organ metastases such as lung, we found marked upregulation of H3K27me3 level compared to matched primary tumors, but EZH2 level on these metastatic sites failed to correlate with H3K27me3 expression suggesting EZH2-independent mechanisms may be involved in such cases. Demethylases like UTX/KDM6A could be potential candidates to be considered for further investigation[27]. Therefore, the clinicians should carefully consider the EZH2 therapeutic window as EPZ6438 (EZH2 inhibitor) or Tazemetostat recently received FDA approval against sarcoma[35,36].

In conclusion, our results reveal that the EZH2 catalytic activity enhances KRT14 expression in the basal-like TNBC subtype. Mechanistically, we have excavated that instead of the classical suppression function, H3K27me3 can promote specific gene expression, like KRT14 transcriptional upregulation, to govern the TNBC peritoneal metastasis. Further, we corroborate the critical involvement of the EZH2-H3K27me3-KRT14 axis in human TNBC metastasis. Finally, we identify that the EZH2 inhibitor drug Tazemetostat (EPZ6438) can be a promising therapeutic option against the most aggressive TNBC subtype, where targeted therapy is still an enigma.

## Methods
### Study approval
All animal studies were conducted by following standard principles and procedures approved by the Institutional Animal Ethics Committee (IAEC) of CSIR-Central Drug Research Institute (Protocol Number: IAEC/2018/F-65). All studies with clinical specimens were approved by the Institutional Review Board of the Rajiv Gandhi Cancer Institute and Research Center (RGCIRC), New Delhi, India (Protocol Number: Res/

BR/TRB-24/2021/43) and CSIR-CDRI Institutional Human Ethics Committee (IEC), Lucknow, India (Protocol Number: CDRI/IEC/2022/A9).

### Reagents and antibodies
Dimethyl Sulfoxide (DMSO), bovine serum albumin (BSA), anti-β-Actin (cat# A3854, 1:10,000) antibody, Crystal violet dye, doxycycline, and Polybrene were purchased from Sigma-Aldrich. EPZ6438 was purchased from Apex biosciences. XenoLight D-Luciferin potassium salt (P/N 122799) obtained from Perkin Elmer. ProLong™ Gold Antifade Mountant was purchased from Invitrogen. Isoflurane (FORANE) was bought from Baxter U.S. Health Care. Matrigel Invasion Chamber (24 well plate 0.8 microns, Lot-8351001) was purchased from Corning. Magnetic ChIP kit and antibodies for EZH2 (cat# 5246S), H3k27me3 (cat# 9733S), H3k4me3 (cat# 9751S), p-PolII-S5 (cat# 13523S), SP1 (cat# 9389S) were purchased from Cell Signaling Technology (CST) and used in 1:1000 dilution for WB studies and 1:50 for IHC and ChIP studies wherever applicable. The antibody for GAPDH (#25778, 1:1000) and anti-mouse SP1 (cat# sc-17824, 1:500) were purchased from Santa Cruz Biotechnology. Antibody for KRT14 (cat# ab7800, 1:1000), and Ki-67 (cat# ab16667, 1:1000) were procured from Abcam. PVDF membrane and stripping buffer were obtained from Millipore Inc. BCA protein estimation kit, RIPA cell lysis buffer, blocking buffer, Super Signal West Pico and Femto chemiluminescent substrate, Lipofectamine-3000, Puromycin, Alexafluor 488/594 conjugated secondary antibodies, FBS, RPMI-1640 media, Anti-Anti, DYNAmo Color Flash SYBR Green qPRC kit (cat# F-416L) were purchased from Thermo Fisher Scientific. Primers for real-time PCR and ChIP assay were purchased from IDT Inc. (Details of primer are listed in Supplementary Tables 2 and 3). A dual luciferase assay kit (cat# E1910) was purchased from Promega. Iscript AdV cDNA kit for RT-PCR (cat# 1725038) was purchased from *BIO-RAD*. RNeasy Mini Kit (cat#74104) was bought from Qiagen. Ex prep™ plasmid SV mini (cat# 101-150) procured from Gene All. The scrather was purchased from SPL life sciences. The collagenase Type I (cat# 1700-017) was purchased from Gibco. All chemicals and antibodies were obtained from Sigma or Thermo scientific unless specified otherwise.

### Procurement and culture of cell lines
Various basal-like or basaloid TNBC human (HCC1806, CRL-2335 and MDA-MB-468, HTB-132) and mouse (4T-1, CRL-2539) cell lines were obtained from American Type Culture Collection (ATCC), USA. Mycoplasma-free early passage cells were revived from liquid nitrogen vapor stocks and inspected microscopically for stable phenotype before use. Human cell lines used in the study are authenticated by STR profiling. All experiments were performed within early passages (<10) of individual cell lines. Cells were cultured as monolayers in recommended media supplemented with 10% FBS, 1-X anti−anti (containing 100 μg/ml streptomycin, 100 unit/ml penicillin, and 0.25 μg/ml amphotericin B) and maintained in 5% $CO_2$ and humidified environmental 37 °C.

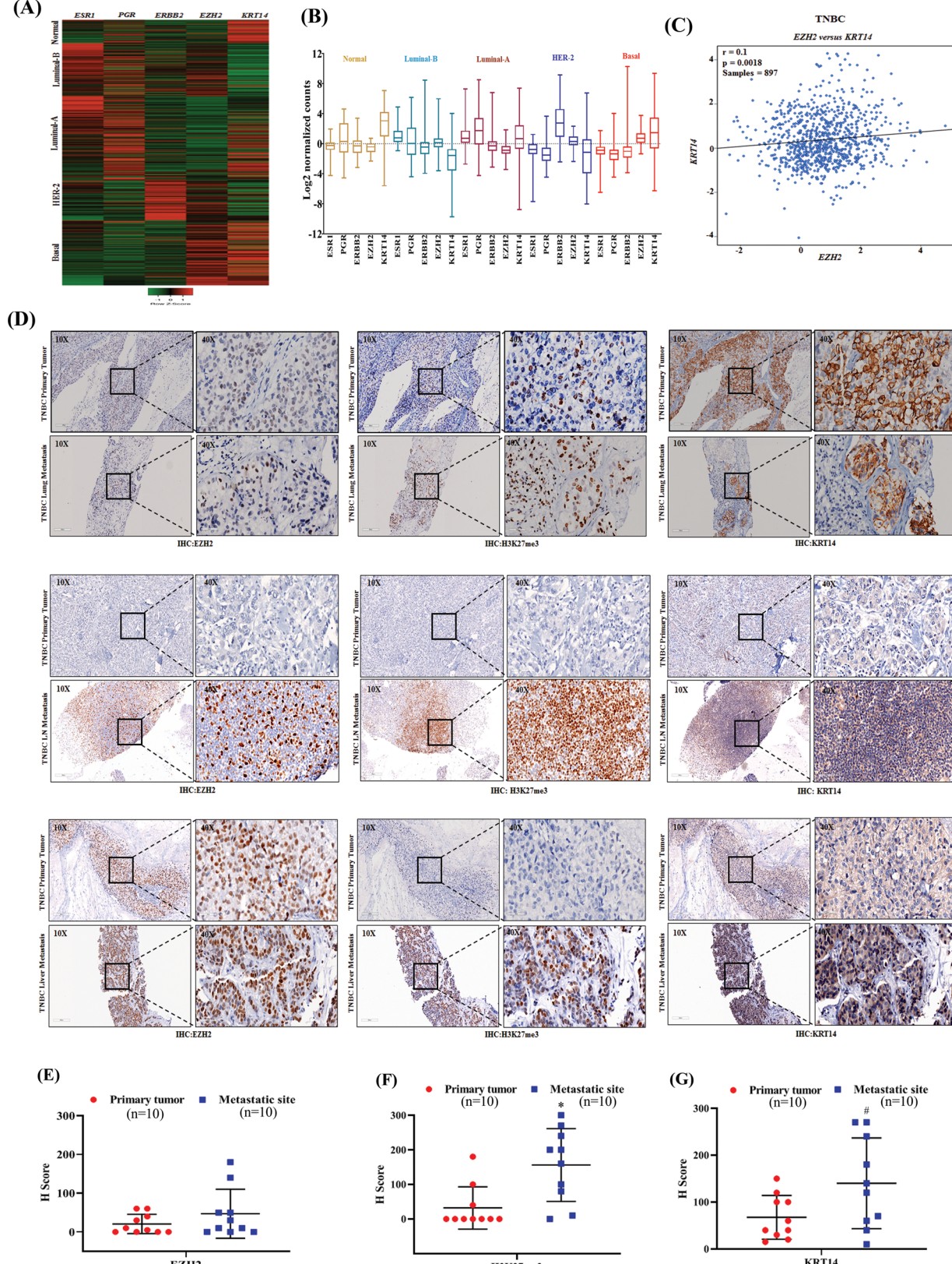

## Generation of stable cell lines

MSCV (cat# 24828), MSCV EZH2 ΔSET-Hygro (cat# 49403) EZH2-Y641-F (cat# 80077), pTRIPZ M)-YFP-EZH2 (cat# 82511), MSCV-EZH2OE (cat#75125), MSCV–EZH2^EZH1SET (cat#75126) were procured from Addgene USA. Control (MSCV), EZH2 ΔSET-Hygro, and EZH2-Y641-F cell lines were generated by utilizing a retroviral-mediated transduction system followed by puromycin selection. The HEK293-T (ATCC- CRL-3216) cell line was used for the generation of viral particles following standard protocol. The HEK293-T cells were plated in the 6-well plate at 80% confluency. Polybrene (8 µg/ml) was added to the viral soup during the transduction of matured viral particles into the target cells. MSCV control cells were subjected to puromycin, and EZH2 Y641-F and

**Fig. 8 | H3K27me3 and KRT14 levels are significantly increased in human TNBC metastasis. A** Analysis of *EZH2* and *KRT14* transcript levels on the basis of ESR1, PGR, and ERBB2 expression in different PAM50 subtypes represented in a heat map. Data are retrieved from the Breast Cancer (Yau 2010) data set, publicly accessible via XENA USSC Cancer Genome Browser. **B** Representative of heat map in the form of box and Whiskers plot for Breast Cancer (Yau 2010) PAM50 subtypes, Normal (Brown) *n* = 66, ESRI s.d. = 1.01, mean = −0.26, PGR s.d. = 2.35, mean = 0.474, ERBB2s.d. = 1.41, mean = −0.212, EZH2 s.d. = 0.80, mean = −0.505 and KRT14 s.d = 2.33, mean = 2.76. Luminal B (sky blue) *n* = 139, ESR1 s.d. = 0.39, mean = 0.071, PGR, s.d. = 2.44, mean = 0.155, ERBB2 s.d. = 1.36, mean = −0.420, EZH2 s.d. =1.20, mean=0.366, KRT14 s.d = 2.47, mean = −2.28.Luminal A n = 222, ESR1 s.d.=0.45, mean = −0.301, PGR s.d. = 2.27, mean=1.26, ERBB2 s.d.=1.43, mean = −0.195, EZH2 s.d. = 0.725, mean = −0647 and KRT14 s.d. = 2.80, mean = 0.752. HER+ *n* = 102, ESRI s.d. = 1.70, mean = −1.842, PGR s.d. = 1.56, mean = −1.48, ERBB2, s.d. = 2.59, mean = 3.29, EZH2 s.d. = 0.78, mean = 0.303 and KRT14 s.d. = 2.79, mean = −1.906. Basal *n* = 170, ESR1 s.d. = 1.80, mean = −2.76, PGR s.d. = 0.791, mean = −1.54, ERBB2 s.d. = 2.50, mean = −1.18, EZH2 s.d. = 0.98,

mean = 1.97 and KRT14 s.d. = 3.42, mean = 1.45. Whiskers for the plot signify SD and the bar denotes the mean for each subtype, where the box extends from 25th to 75th percentile, and whiskers range from minimum and maximum value, with the center denoting the median value. **C** Correlation plot showing IHC co-expression of the EZH2 and KRT14 in breast cancer TNBC subtype. Pearson pairwise correlation coefficient on 897 TNBC samples shows a positive correlation (*r* = 0.10) between EZH2 and KRT14 and is significantly associated with each other (*p* = 0.0018). **D** Immunohistochemistry was carried out to detect EZH2, H3K27me3, and KRT14 in FFPE serial sections of matched human TNBC primary tumor and respective metastatic counterparts using anti-EZH2, anti-H3K27me3, and anti-KRT14 antibodies. Representative photomicrographs were shown at 10X and 40X magnifications (inset). Scale bar, 200 μm (10×) or 50 μm (40×). **E**–**G** Quantitative H-scores for TNBC primary tumors and metastatic counterparts (*n* = 10) were calculated for EZH2 (**E**), H3K27me3 (**F**), and KRT14 (**G**) expression and represented as scatter plots; error bar, ±SD, left to right *P = 0.0047, #P = 0.046, compared to expression in respective primary tumors, Student's *t*-test (two-sided). Source data are provided as a Source Data file.

EZH2 ΔSET cells were subjected to hygromycin selection, and the overexpression of stable EZH2, ΔSET, and EZH2-Y641-F was confirmed by western blot. Following the same protocol reported in our recent publication[66] *EZH2* mouse, *KRT14* mouse, and Human shRNA, SP1 mouse, and human and *UTX* mouse shRNA sequence were cloned in the 3rd generation transfer plasmid pLKO.1 TRC cloning vector (Addgene cat # 10878) between unique AgeI and EcoRI restriction sites downstream of the U6 promoter. HEK-293T cell line was used for the generation of lentiviral particles, and media containing the viral particles was supplemented with Polybrene (8 μg/ml) for the transduction purpose. Cells were subjected to puromycin selection after 48 h of transduction, and the knockdown for EZH2 and KRT14 was confirmed by western blot. shRNA sequences were listed in Supplementary Table 4.

## Patient sample collection

A total of 3500 Breast Cancer patients were screened that had reported to RGCIRC between the years 2015–2019. The time frame chosen was such that a 2-year follow-up period could be accounted for. Out of these 3500 patients, 600 cases qualified as TNBC patients based on IHC and Her2 by FISH analysis and had also undergone surgery and other treatments at RGCIRC. Complete follow-up data of these 600 patients were analyzed to look for disease progression in the form of local recurrence or distant metastasis. Approximately, 60 patients had recurred or progressed at varying time points. Of these 60 cases, 10 TNBC cases (patient details are enlisted in Supplementary Table 6) were selected wherein there was a matched tumor block, and a matched metastatic site block was available. Formalin-fixed, paraffin-embedded specimens of 10 primary TNBC cancer tissues and matched 10 metastatic samples were used for IHC staining.

## Immunohistochemical (IHC) staining and analysis

IHC staining was performed using Ventana Benchmark XT automated closed system following their recommended protocol. EZH2 and H3K27me3 antibodies were purchased from Cell Signaling Technology, and KRT14 was obtained from Abcam. Colon, Skin, and Appendix were used as internal positive control for KRT14, EZH2, and H3K27me3 staining, respectively. All stained slides were analyzed by Leica Aperio ImageScope software. The staining intensity of each section was scored as 0 (no staining), 1+ (weak staining), 2+ (moderate staining), or 3+ (strong staining). The tumor cell positive rate (0–100%) per slice was multiplied by the staining intensity to get an overall H-scores ranging from 0 to 300.

## Cloning of *KRT14* promoter in luciferase reporter vector and luciferase assay

The 500 bp (−0.5 kb from TSS) and 1100 bp (−1.1 kb from TSS) *KRT14* fragments were amplified from Hela cell DNA by PCR. The amplified

fragments were cloned into the PGL4.12 [*luc2 CP*] vector between the HindIII and BglII restriction sites. The control and SP1 KD HCC1806 cells were seeded at 50–60% confluence in 6-well plates and transfected with 5 μg of PGL4-1.1 kb KRT14-P, and 50 ng of PGL4 (hRluc-CMV) plasmid using lipofectamine 3000 as transfection reagent (Invitrogen). The same process was used for PGL4-0.5 kb KRT14-P. For Mithramycin treatment, the PGL4-0.5 kb KRT14-P transfected HCC1806 cells were treated either with vehicle control or two doses of Mithramycin (150 and 300 nM) for 24 h. Lysis buffer contained in the Dual-Glo Luciferase assay kit (Promega) was used to lyse the cells. The activities of Firefly and Renilla luciferases were measured using the GloMax® 96 Microplate Luminometer according to the manufacturer's procedure (Promega). Firefly luciferase activity was normalized to Renilla luciferase activity for each sample. Luciferase KRT14 promoter primers are enlisted in Supplementary Table 5.

## MTS assay

The 4T-1 cells were seeded in the 96-well plates at a density of 8000 cells per well in the 5% serum-containing media (100 μl) overnight. The adhered cells were treated with vehicle control and 3 doses of EPZ6438 (10, 20, and 40 μM) for 48 h. for MTS assay, the Cell Titer96 R Aqueous One Solution Proliferation Assay Kit was used following the manufacturer's instruction. The 20 μl of the MTS reagent was added to each well and incubated at 37 °C for 4 h. The absorbance was detected at 490 nm by the plate reader.

## Western blotting

Cells were subjected to lysis in RIPA buffer containing phosphatase and protease inhibitor cocktail and incubated at −20 °C for 48 h subsequently, the samples were thawed at RT and centrifuged at 5000*g* for 15 min at 4 °C, the supernatant was collected, and the pallet was discarded. The Protein concentrations were estimated by utilizing the BCA kit. Equal amounts of protein were resolved by SDS-PAGE and transferred to a PVDF membrane. Membranes were blocked with 5% nonfat dry milk or 5% BSA, followed by incubation with appropriate dilutions (1:1000) of primary antibodies overnight at 4 °C and subsequently incubated with a 1:5000 dilution of horseradish peroxidase-conjugated secondary antibodies for 1 h at room temperature. Immunoreactivity was detected by enhanced chemiluminescence solution (ImmobilonTM western, Millipore, USA) and scanned by the gel documentation system (Bio-Rad chemidoc XRS plus).

## RNA Seq analysis

Illumina NovaSeq was used to perform transcriptome sequencing, followed by the Fast QC and Multi QC tools were used to assess data quality. Adapter sequences (P7 adapter read 1: AGATCGGAAGAGCA

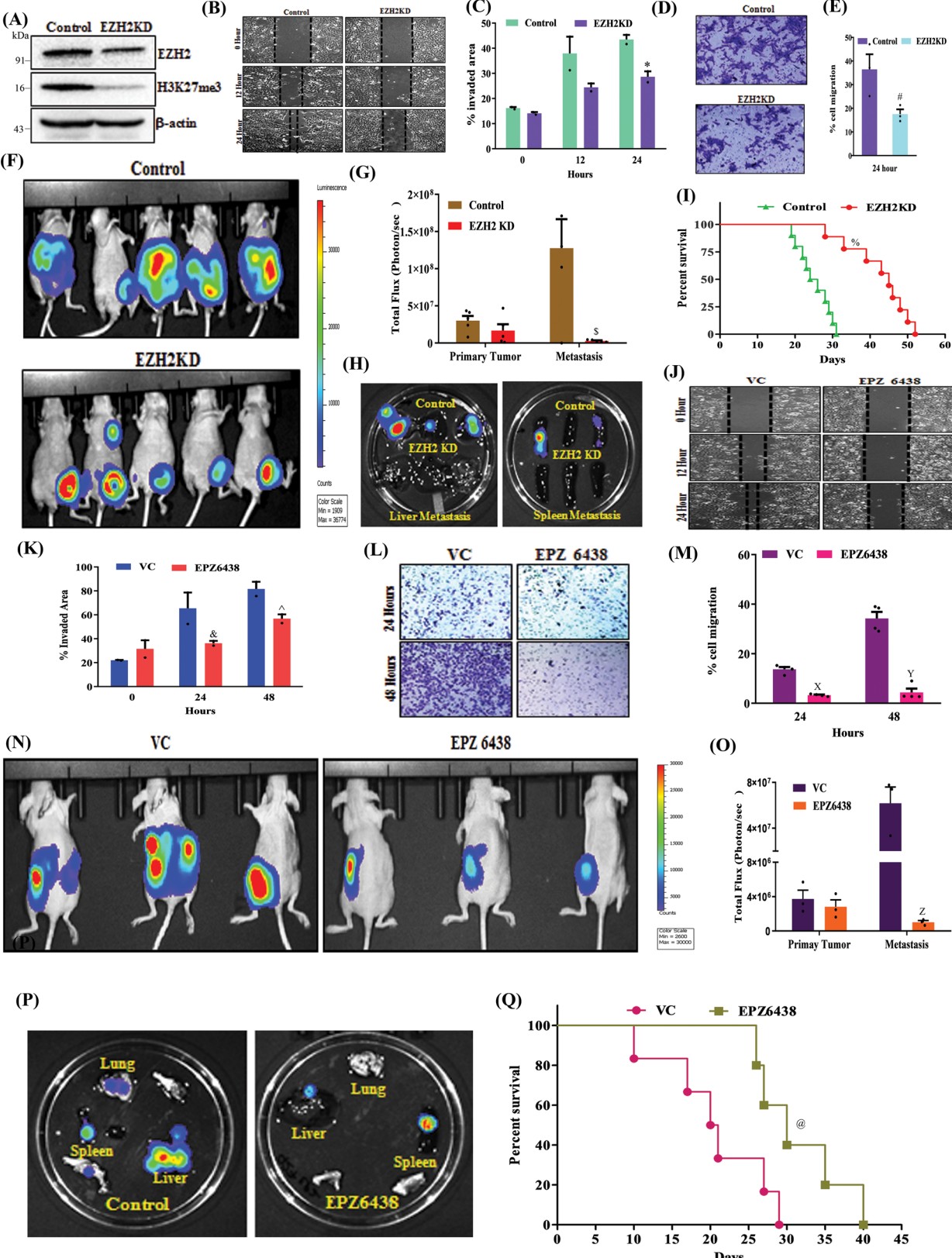

CACGTCTGAACTCCAGTCA and P5 adapter read 2: AGATCGGAAG AGCGTCGTGTAGGGAAAGAGTGT) and low-quality bases were removed from raw sequence reads using fastp. Using the STAR v2 aligner, the QC-passed reads were mapped onto the indexed *mus musculus* genome (GRCm39)[67]. Gene expression levels were calculated as read counts using feature-counts software. Spearman Rank correlation and principal components analysis (PCA) was used to assess the similarity of expression between biological replicates. For differential expression analysis, the biological replicates were grouped as Control and Y641-F. Differential expression analysis was carried out using the edgeR package after normalizing the data based on trimmed mean of M (TMM) values[68]. After normalization, 19162 features

**Fig. 9 | Inhibition of EZH2 methyltransferase activity reduces TNBC peritoneal metastasis. A** Immunoblot for EZH2, H3K27me3, and β-actin expression in Control and EZH2 knockdown (KD) 4T-1 cells. **B–E** Representative images of the wound healing (**B**) and trans-well chamber migration (**D**) assays in control and EZH2 KD 4T-1 cells, 10× magnification. The quantitative analysis of wound healing (**C**) and trans-well (**E**) assay; Columns, mean of triplicate readings of samples, error bar, ±SD, compared to control, ˙$P = 0.0453$, two-way ANOVA, Sidak's multiple comparisons test and #$P = 0.0467$ student's *t*-test (two-sided). **F** Bioluminescence image of tumor-bearing control (top panel), EZH2 KD (bottom panel) mice. The color scale indicated the photon flux (photon/s) emitted from each group. **G** Quantitative bar graph representation of total photon flux calculated from the region of flux (ROI). Columns, a mean of quadruplicate readings of samples, error bar, ±SD, $^{\$}P = 0.0010$ compared to control, two-way ANOVA, Sidak's multiple comparisons tests. **H** Bioluminescence images of liver and spleen harvested from control and EZH2 KD tumor-bearing mice. **I** The Kaplan–Meier survival curve of control and EZH2 KD 4T-1 tumor-bearing mice cells ($n = 10$), $^{\%}P = 0.0001$, compared to control, log-rank test.

**J–M** Representative images of the wound healing (**J**) and trans-well chamber migration (**L**) assays in 4T-1 EZH2Y641-F (control) and EPZ6438 (10 μM) treated cells, 10× magnification. The quantitative analysis of wound healing (**K**) and trans-well (**M**) assay; Columns, a mean of duplicate (**K**) and triplicate (**M**) readings of samples, error bar, ±SD, compared to control, $^{\&}P = 0.0158$, ˙$P = 0.0219$, $^{X}P = 0.0011$, $^{Y}P = 0.0001$, two-way ANOVA, Sidak's multiple comparisons test. **N** Representative bioluminescence images of EZH2 (Y641-F) 4T-1 tumor-bearing mice, treated with either vehicle or EPZ6438 (250 mg/Kg). **O** Quantitative bar graph representation of total photon flux calculated from the region of flux (ROI); columns, a mean reading of triplicate samples, error bar, ±SD, $^{Z}P = 0.0078$, compared to control, two-way ANOVA, Sidak's multiple comparisons test. **P** The bioluminescence analysis of the metastatic signal in the organs harvested from control (Y641-F) and EPZ6438 treated mice. **Q** The Kaplan–Meier survival curve of nude mice, either treated with vehicle control ($n = 5$) or EPZ6438 (250 mg/kg) ($n = 5$). $^{@}P = 0.0326$, compared to vehicle control, log-rank test. Source data are provided as a Source Data file.

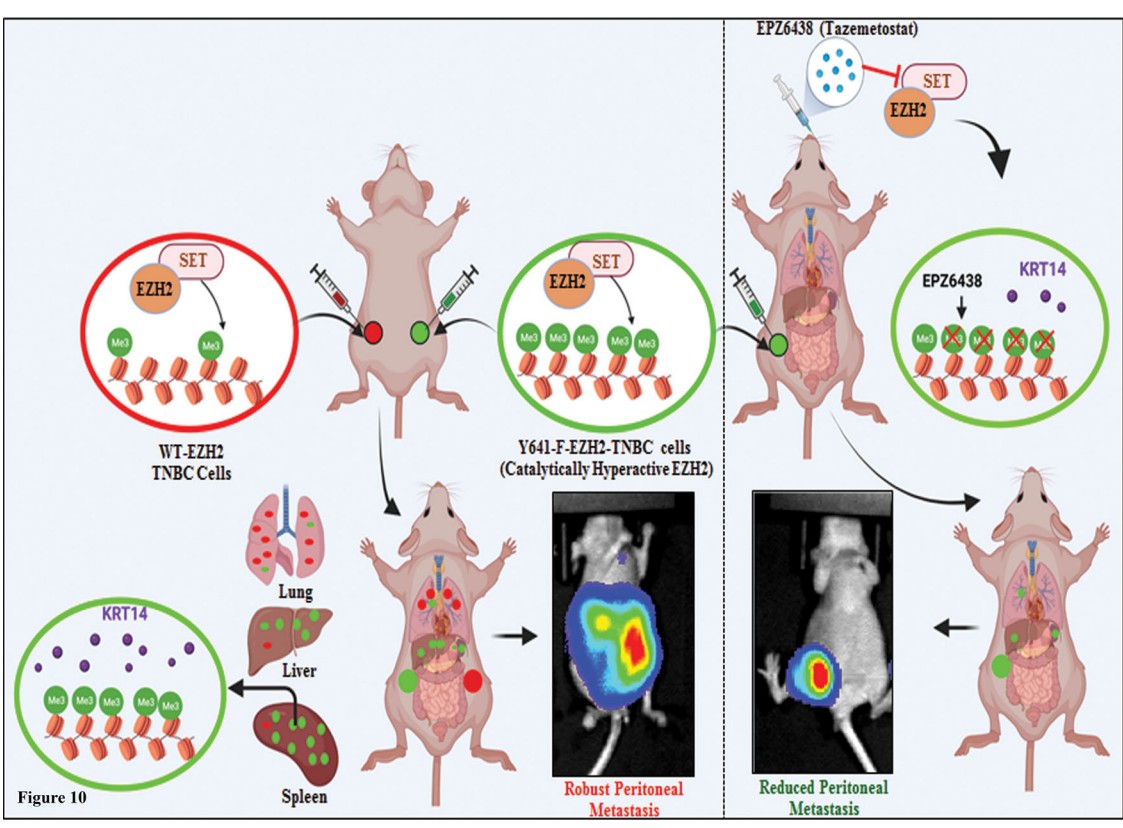

**Fig. 10 | EZH2-H3K27me3 in TNBC metastasis.** Schematic representation of how catalytically hyperactive EZH2 (increased H3K27me3) can promote TNBC peritoneal metastasis and its therapeutic vulnerabilities against EZH2 inhibitor EPZ6438.

(47.66%) were removed from the analysis because they did not have at least 0.1 counts per million in three replicate samples. Genes with absolute Log$_2$FC + 1, −1 and *p*-value ≤ 0.05 was considered significant. The expression profile of differentially expressed genes across the samples is presented in the volcano plot.

## Gene set enrichment analysis

GSEA was performed using the updated version GSEA 4.2.2 (https://www.gsea-msigdb.org/gsea/index.jsp) using h.all.v7.5.1.symbols.gmt (Hallmarks) gene set database and mouse gene symbol remapping human orthologs MSigDB v7.5.1.chip with allowed 1000 permutations and permutation type selected was phenotype[69].The heat map of the genes enriched at the top or bottom of the gene sets were identified using the FDR value threshold ($p < 0.05$) and ranked according to the normalized enrichment score.

## Real-time PCR

Total RNA was isolated from cultured cells using the standard procedure of the RNeasy Mini Kit (Qiagen, cat # 74104). The concentration and purity of the RNA samples were determined using nanodrop. The total RNA (7 μg) of each sample was reverse-transcribed (RT) with iscript ADV cDNA synthesis kit. The final cDNA was diluted with nuclease-free water (1:3), 1 μl of this having a concentration of 80 ng/μl was used for each reaction in real-time PCR. Real-time PCR was carried out using an ABI Step One Plus Real-Time PCR System (Applied Biosystems). Reactions for each sample were performed in triplicate. 18 s amplification was used as the housekeeping gene. A gene expression score was calculated by taking two raises to the difference in Ct between the housekeeping gene and the gene of interest (2 ΔCt). For amplification of *NIFK, ADAMTS1, CCN2, JAG1, SEMA3C, TM4SF1, CYP1B1, KRT14, KRT16, NCAM1, AQP1, BNIP3, CBS, CCNG2, ARRB1, NDRG1, and*

SP1, we performed SYBR Green-based RT-PCR following manufacturer's instructions. Primers were listed in Supplementary Table 2.

### Chromatin immunoprecipitation (ChIP) assay

ChIP assay was conducted by using the ChIP assay kit (Cell Signaling Technology) following the manufacturer's protocol. In brief, cells at 80% confluence were fixed with formaldehyde (1% final concentration directly to the culture media) for 10 min. Cells were then centrifuged, followed by lysis in 200 μl of membrane extraction buffer containing protease inhibitor cocktail. The cell lysates were digested with MNase for 30 min at 37 °C to get chromatin fragments followed by sonication (with 20 s on/20 s off 3 Sonication cycles at 50% amplitude) to generate 100–500 bp long DNA fragments. After centrifugation, clear supernatant was diluted (100:400) in 1× ChIP buffer with protease inhibitor cocktail followed by keeping 5% of input control apart and incubated with primary antibody or respective normal IgG antibody overnight at 4 °C on a rotor. The next day, IP reactions were incubated for 2 h in ChIP-Grade Protein G Magnetic Beads, followed by precipitation of beads and sequential washing with a low and high salt solution. Then elution of chromatin from Antibody/Protein G Magnetic beads and reversal of cross-linking was carried out by the heat. DNA was purified by using spin columns, and SYBR Green-based real-time PCR was conducted. Primer sequences used for the ChIP experiment for different genes are enlisted in Supplementary Table 3.

### Confocal microscopy

Control and treated cells were fixed with ice-cold pure methanol for 10 min at −20 °C, followed by blocking with 2% BSA for 1 h at RT. After overnight primary antibodies (anti-H3K27me3 and anti-KRT14) incubation, cells were washed twice with PBS and incubated with fluorescent-conjugated secondary antibodies at RT for 1 h, followed by DAPI staining for 5 min at RT. After washing, cells were mounted with the anti-fade mounting medium on glass slides and viewed under an inverted confocal laser scanning microscope (Zeiss Meta 510 LSM; Carl Zeiss, Jena, Germany). Plan Apochromat63×/1.4NA Oil DIC objective lens was used for imaging and data collection. Appropriate excitation lines, excitation, and emission filters were used for imaging.

### Wound healing assay

For the wound healing assay, 500,000 cells were seeded into 6-well plates and incubated for overnight to form a confluent monolayer. The scratch has been made by scrather to generate a straight-line scratch in the cell monolayer. The cells were washed with PBS and cultured with fresh complete RPMI media at 37 °C for different time intervals. The cells were incubated at 37 °C for different time intervals with or without treatments. Five (5) reference points were randomly selected from a single well at different time intervals, and the percentage of wounds healed area was measured by Image J software. Three independent replicate experiments were conducted for single data representation.

### Invasion assay

For each invasion assay, cells were re-suspended in 500 μl of serum-free RPMI and were added to the inside of Matrigel inserts (Coring Bio Coat), and DMEM media with the 10% serum was added outsides of inserts. The Matrigel invasion chambers were incubated for 24 h at 37 °C. The non-invading cells were then removed by scrubbing with a cotton-tipped swab. Invasion chambers were fixed with 100% methanol for 5 minutes. The invasion chambers were stained with crystal violet for 1 h and then washed with PBS twice. Five reference points were randomly selected for each invasion camber. The number of invaded cells was analyzed by Image J software. Three independent replicate experiments were conducted for single data representation.

### Animal studies

Experimental mice were maintained in IVC cages under pathogen-free conditions with a 12 h light/12 h dark cycle, at 24 ± 2 °C temperature with humidity of 45 ± 5%, and were fed with irradiated standard mouse diet at CSIR-CDRI Central Laboratory Animal facility. Six-week-old female Balb/c nude mice were used for all studies. For orthotopic inoculation, different tagged (GFP, Td-Tomato, Luc) 4T-1 cells ($1 \times 10^6$) in 100 μl were injected into the mammary fat pad of 4–6-week-old nude Crl: CD1-Foxn1$^{nu}$ female mice, whereas, the same number of cells were inoculated in dorsal right flank for subcutaneous mice model. For tracking of genetically manipulated metastatic cells, as described in Fig. 4A–C, the following strategies were adapted. First strategy: The Td-Tomato$^+$ control and EZH2 (Y641-F) GFP$^+$ cells were mixed in an equal ratio (1:1). The 100 μl of mixed cells ($0.5 \times 10^6$ Td-Tomato$^+$ $0.5 \times 10^6$ Y641-F-GFP$^+$) were orthotopically inoculated in the right mammary fat pad of female nude ($n = 5$). Second strategy: The MSCV Td-Tomato$^+$ control and EZH2 (Y641-F) GFP$^+$ cells were mixed in an equal ratio (1:1). The 100 μl of mixed cells ($0.5 \times 10^6$ Td-Tomato$^+$ $0.5 \times 10^6$ Y641-F–GFP$^+$) were inoculated via tail vein in the female nude mice ($n = 5$). Third strategy: the control ($0.5 \times 10^6$ Td-Tomato$^+$) and EZH2 ($0.5 \times 10^6$ Y641-F GFP$^+$) cells in 100 μl PBS were orthotopically inoculated in the right and left mammary fat pad of female nude mice ($n = 5$). EPZ6438 (250 mg/kg dose) or vehicle (0.5% NaCMC + 0.1% Tween-80 in water) was administered per day by oral gavage for 24 days after 1 week of post tumor cell inoculation. Throughout the study, tumors were measured with an electronic digital caliper at regular intervals, and the tumor volume was calculated using the standard formula $V = \Pi/6 \times a^2 \times b$ ('a' is the short and 'b' is the long tumor axis). At the end of the experiment, mice were sacrificed, and tumors were dissected for further studies. The tumor volume of mice did not go beyond 2500 mm$^3$ as allowed by the Institutional Animal Care and Ethical Committee. Live animal bioluminescent imaging (IVIS spectrum, Perkin Elmer) was performed once per week of post-inoculation. For in vivo imaging studies, 150 mg/kg D-Luciferin (10 mg/ml in PBS) was injected intraperitoneal in the tumor-bearing mice. Subsequently, mice were anesthetized by Isoflurane. Images were captured with dorsal and ventral positions using Perkin Elmer IVIS system coupled with bioluminescence image acquisition and analysis software. Regions of interest (ROI) from displayed images were identified on the tumor and metastatic sites and quantified as photons per second (p/s) using Living Image software. Spectral unmixing was used to detect Td–Tomato and GFP signals in the same mice, and finally, data were acquired at 570 nm (Td-Tomato) and 465 nm (GFP).

### Colony formation assay

The 4T-1 cells were seeded at 200 cells per well in 12-well plates, after 24 h, the adhered cells were treated either with the vehicle and different doses of EPZ6438 (10, 20, and 40 μM) and incubated for a week at 37 °C. Subsequently, the media was aspirated, followed by two times PBS washing. The washed cells were fixed with ice-cold methanol for 10 min and stained with 0.5% crystal violet dye for 1 h. The excess stain was washed with water, and the plates were allowed to air dry. The stained colonies were counted by Image J Software.

### Tumor cell isolation from peritoneal fluid

4T-1 tumor-bearing mice were euthanized and sprayed with 70% ethanol. The outer skin of the peritoneum was cut with the help of scissors and forceps, and then the skin was gently pulled back to expose the inner skin, lining the peritoneal cavity. Ice-cold PBS was injected into the peritoneal cavity with the help of a needle, and the peritoneal fluid with cell suspension was collected and kept in tubes on ice. The collected cell suspension was spun at 1500 RPM for 10 min, the supernatant was discarded, and cells were resuspended in desired media in the presence of 60 μM of 6-TG.

## Isolation, culture, and analysis of metastatic cells

Mice were sacrificed, and primary tumors and other organs were harvested under sterile conditions. Single-cell suspension was prepared following standard protocol. Briefly, chopped tissues were incubated in HBBS solution containing 1 mg/ml of Collagenase on a rocker for 2 hr at 37 °C. The suspension was then centrifuged for 5 min at 500×$g$, washed, and the cell pellet was re-suspended in normal growth medium (RPMI, 10% FBS, 1× anti-anti) onto a T-25 flask. After 24 h, fresh media containing 60 µM 6-Thioguanine was added and cultured for 3 days to keep only 6-Tg resistant 4T-1 cells. Isolated cells from primary tumors and metastatic organs were used for further analysis.

## Analysis of the breast cancer dataset

Illumina HiSeq mRNA data of patients with Breast cancer (Yau 2010) was downloaded from UCSC Xena[70] for *ESR1*, *PGR*, *ERBB2 EZH2*, and *KRT14*. The breast cancer dataset was segregated into a PAM50 subtype for luminal A, Luminal B, HER2+, basal and normal-like subtypes, and a heatmap was generated.

## Statistics and reproducibility

Data are presented as mean ± SEM or mean ± SD, as indicated, of at least three independent experiments or biological replicates. Two-way ANOVA, one-way ANOVA, Student's $t$-test, and two-tailed distributions were used to calculate the statistical significance of in vitro and in vivo experiments. The Kaplan–Meier survival curve significance was analyzed by log-rank test. These analyses were done with GraphPad Prism software. Results were considered statistically significant when $p$-values ≤ 0.05 between groups. All data shown are representative of three independent experiments with similar results unless otherwise indicated.

## Reporting summary

Further information on research design is available in the Nature Portfolio Reporting Summary linked to this article.

## Data availability

Data were curated using publicly available databases such as UCSC Xena online Browser (https://ucsc-public-main-xena-hub.s3.us-east-1.amazonaws.com/download/YauClinical_public%2FYauGeneExp_genomicMatrix.gz; Full metadata) Breast Cancer Gene Expression Minor Version V 4.8 (http://bcgenex.ico.unicancer.fr/BC-GEM/GEM-requete.php), Eukaryotic Promoter Database, TRRUST (Version 2), JASPAR (https://jaspar.genereg.net/). The RNA-seq data relevant to this study have been deposited in the Gene Expression Omnibus database under accession code GSE217474. All data needed to evaluate the conclusions in the paper are present in the paper and/or in the Supplementary Information. Additional data related to this paper is available in 'Source Data' File. Source data are provided in this paper.

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

## Acknowledgements

We sincerely acknowledge the excellent technical help of Mr. A.L. Vishwakarma of SAIF for the Flow Cytometry studies; Ms. Reema of the Electron Microscopy unit for Confocal Imaging. We express our deepest gratitude to Dr. S.K. Rath and Mr. Navadayam for providing the Imaging facility. The authors are immensely grateful to Dr. Juhi Tayal, Dr. Anurag Mehta, Dr. D.C. Doval, and Ms. Somika Tiwari (Biorepository, Rajiv Gandhi Cancer Institute, and Research Center, New Delhi, India) for TNBC tumor samples and their excellent technical support for IHC studies. We extend our heartfelt gratitude to Dr. Perumal Nagarajan and Dr. Jayanta Sarkar for their immense guidance in maintaining immunocompromised animal facilities. Research of all the authors' laboratories was supported by CSIR Pan-Cancer Grant (HCP-40 to D.D., M.P.S.) and Fellowship grants from CSIR (A.K.S., K.K.S.), DBT (P.C.), and UGC (A.V.). Further, D.D. acknowledges grant support from CSIR-FTT (MLP-2025), CSIR-FBR (MLP-2027), DST (EMR/2016/006935), DBT (BT/AIRO568/PACE-15/18), and ICMR (2019-1350). Diagrammatic Figures

were created using BioRender software (https://biorender.com), and D.D. owns a full license to publish. We dedicate this paper to Prof. Tushar Kanti Chakraborty, who extended selfless support in establishing PI's independent laboratory. The Institutional (CSIR-CDRI) communication number for this article is 10504.

## Author contributions

A.V. was involved with study designing, performed experiments, and wrote the draft paper. AS helped in carrying out in vivo studies. M.P.S. performed bioinformatic analysis. M.A.N., K.K.S., A.B.S., P.C., S.R.S., M.A.K., S.M., and A.K.S. provided active support for carrying out various in vitro and in vivo experiments. D.D. conceived the idea, designed experiments, analyzed data, wrote the paper, and provided overall supervision. All authors read and approved the final paper.

## Competing interests

The authors declare no competing interests.
