## [Peer Review File · Nature Communications]

Reviewer comments, first round review -

Reviewer #1, expert in EZH2 and breast cancer metastasis (Remarks to the Author):

This is an interesting study on the importance EZH2 enzymatic activity on TNBC metastasis and a novel mechanism. There are a few comments that need to be addressed.

1. The investigators compare a SET deletion mutant (truncated EZH2) with a point mutation Y-641-F to make strong conclusions. The dSET mutant, as it is a deletion mutant, may have unexpected functions that are not solely due to the lack of enzymatic activity. Another approach is necessary to validate the results obtained with dSET. Also, while an activating mutation of EZH2 is a good approach, at least another approach to enhance EZH2 H3K27me3 activity should be used to validate the results.
2. The in vitro studies in Figure 1 only compare Y-641-F to control, no experiments with dSET or EZH2 enzymatic activity inhibition are shown.
3. 4T1 cells are aggressive breast cancer cells with already high levels of EZH2 expression. May not be the best model to further increase EZH2 activity.
4. Further explanation on how genes were "manually selected" for validation is needed.
5. There are defined targets of EZH2 in breast and other cancers (e.g. DKK1, hox genes). Were these genes present in this study?
6. In Figure 7, how are peritoneal metastasis defined? There are no histological images to show cancer cells in the peritoneum or in peritoneal fluid.
7. The correlation shown between EZH2 and KRT14 is through data mining of transcript expression (7R). There is no demonstration that EZH2 and KRT14 proteins are concordantly expressed in breast cancer, and their association in TNBC and TNBC metastasis.
8. EZH2 KD has been shown previously to reduce invasion, migration, and other tumorigenic functions, and needs to be referenced appropriately (Moore HM. Breast Cancer Res Treat 2013 138:741-52, Anwar T Nat Commun 2018 9:2801)

Reviewer #2, expert in breast cancer metastasis (Remarks to the Author):

In this MS, the authors investigated the underlying mechanism of basal-like TNBC peritoneal metastasis. They found EZH2 catalyzed H3K27me3 modification of KRT14 promoter region prevented the binding of SP1 transcriptional factor and therefore relieved the transcriptional repression of KRT14. Inhibition of EZH2 or KRT14 significantly blocked tumor metastasis and increase overall survival of experimental animals.

In general, this MS provided a potential therapeutic target for TNBC peritoneal metastasis, which is a rare and challenging clinical presentation. However, the main conclusion that EZH2-mediated H3K27me3 promoted breast cancer metastasis was not new (Nat Commun. 2018 Jun 29;9(1):2547. Nat Commun. 2018 Jul 18;9(1):2801.). Besides, lack of clinical data to support the conclusion will weaken the data solidarity. Detailed comments list below.

Major comments,

1. The authors should collect clinical samples to observe the correlation between EZH2-H3K27me3-KRT14 level and tumor distant metastasis, especially on sites of peritoneum, and abdominal organs, such as liver and spleen.
2. In the metastasis models, the authors used 4T-1 (Y641-F) and HCC1806 cells. Why HCC1806 cells were not transfected with Y641-F? Does HCC1806 cells have higher H3K27me3 modification?
3. In the animal experiments, the authors should examine ki67, H3K27me3-KRT14 expression (IF blotted as figure 3N) both in primary tumors and metastases to confirm the in vitro results.

Minor comments,

1. What did "control" group mean, cells overexpressed with wt EZH2 or empty vector? In figure 1B, 3H the expression of EZH2 didn't changed a lot in Y641-F group compared to control, had control group also been transfected with similar amount of EZH2?
2. In figure 1C-E, we can see Y641-F significantly decreased tumor volume, however, there were

no significance marks. At the same time, in sup figure 1, it seems Y641-F group had the similar tumor volume with the other two groups. These results were not consistent. The authors should check the data and explain the results.

3. In figure 1G, it's not clear why all 4T1 Y641-F OE tumors had metastasis to the contralateral body. How to explain the route of metastasis?

4. In figure 4B-E, the authors better provide ChIP-seq results to show the broad peaks of H3K27me3 binding in the upstream promoter region of KRT14. So we can easily tell the enrichment region of H3K27me3 in the promoter, and how Y641-F affect H3K27me3 intensity.

5. To show the correlation between EZH2 and KRT14 expression, the authors better provide the correlation analysis.

6. In figure 2E, 3N, 6K, the authors should provide scale bars.

7. In line 357, it should be Figure 7N-7P; and in line 361, it should be Figure 7R-7S.

Reviewer #3, expert in breast cancer transcriptomics (Remarks to the Author):

"H3K27me3 mediated KRT14 upregulation promotes TNBC peritoneal metastasis" by Verma et al. uses in vitro and in vivo modeling to show that splenic metastasis of triple-negative breast cancer (TNBC) is governed by elevated ("hyperactivated") H3K27me3 via the increased catalytic activity of EZH2. EZH2 has been characterized as an oncogene and is overexpressed in many different cancer types. It is a catalytic subunit of PRC2 and functions with EED and SUZ12 as a histone methyltransferase to silence active transcription through deposition of H3K27me3. Verma et al. focus on the catalytic function of EZH2 in basal-like TNBC, although the authors note this has already been described in part by several publications (including a recent study of the Mittal lab, Yomtoubian et al., 2020). The novelty of this work comes from the identification of KRT14, a known marker of TNBC, to be transcriptionally activated by EZH2. Although the therapeutic potential of EZH2 inhibitors in TNBC has been proposed previously (Yomtoubian et al., 2020), the mechanism of action described here is different. Overall, this is an interesting study that finds a role for increased H3K27me3 in peritoneal metastasis in TNBC.

Summary: The authors demonstrate that increased ("hyperactive") H3K27me3 promotes TNBC migration in mice using in vivo and in vitro studies. To do this the authors engineered 4T-1 stable cells with catalytically inactive EZH2 (NC-EZH2) or catalytically hyperactive Y641F EZH2 protein. These genetically modified 4T-1 and control cells were implanted subcutaneously and orthotopically into mice. While NC-EZH2 tumor sizes did not change relative to control, Y641F tumors appeared smaller and Y641F tumor bearing mice lost significant body weight compared to control. Live animal imaging was used to assess the metastatic potential of groups of cells and found that Y641F mice had significant increase in metastasis compared to control. A wound healing assay and trans-well chamber confirmed Y641F mutant cells had higher migratory and invasive potential than WT.

The pattern of metastatic spread was assessed by fluorescence imaging of harvested organs. Control cells tended to spread to lungs whereas, Y641 cells tended to colocalize to the spleen and liver. Splenic cells had highest expression of H3K27me3. Animal imaging of inoculated splenic metastatic cells versus primary tumor cells show splenic cells result in profound increase of TNBC peritoneal metastasis.

RNA-sequencing identified KRT14 among others as significantly upregulated in Y641F cells. Earlier reports corroborate KRT14 as a marker of TNBC basal subtype. Here the authors validated KRT14 mRNA expression and protein levels in Y641F cells and show KRT14 expression is upregulated in Y641F compared to control and to cells treated with tazemetostat an EZH2 inhibitor. To investigate whether H3K27me3 can positively regulate KRT14 they perform ChIP assay and find reduction of H3K27me3 marks, but increased H3K4me3 marks around KRT14 promoter. The authors investigate SP1 as a transcriptional regulator of KRT14, identify its binding motif in the promoter of KRT14 and test association with knockdown. They suggest H3K27me3 may compact the GC box region in the KRT14 promoter to inhibit SP1 binding and allow transcription of KRT14. EZH2 knockdown found reduced peritoneal metastasis compared to control and increased survival in mice. Though an FDA approved EZH2 inhibitor tazemetostat for sarcomas reduced TNBC peritoneal

metastasis, the primary tumor burden was the same compared to control.

Several points need clarification:

1. Figures 1C and 1E should show the statistical significance of comparisons in tumor volume and weight, respectively.
2. The Figure 3B volcano plot shows significant differential expression analysis in just 284 genes (92 up, and 192 down at $p < 0.05$ and $fc \geq 2$). There are few genes here; a table of the gene expression and differential statistics between control and Y641F in the RNA-sequencing data should be provided. Are the p-values in DE analysis FDR-corrected?
3. Figure S2A should be sorted according to significance similarly to Figure 2B.
4. Caption for Figure S2A is KEGG pathway analysis of candidate genes and indicates ($p < 0.5$) was considered for "statically significantly difference". There are typos in the quotes and is the written p-value cutoff ($p < 0.5$) accurate?
5. On page 7 (line 174) it is stated "we extensively searched literature for metastasis related genes..." A little more detail how these genes were 'selected'.
6. Figure 3C heatmap appears to have red as low and blue high. The opposite is more common (red high, blue low), style issue.
7. Neither S2A nor S2B appear to show significant enrichment of metastasis related pathways, yet these were specifically identified in the manuscript (line 176 and heatmap Figure 3C).
8. Line 179 is unclear and should be reworded: "...differentially expressed genes to identify the up and down regulated metastasis related KEGG pathways and most of the manually selected genes, we found common in the metastasis associated pathways. Additionally, the authors should clarify what are "most of the manually selected genes". A pathway analysis using an unbiased approach, such as GSEA that includes expression of all genes between control and Y641F would clarify this analysis.
9. There are 9 lines 178-186 that are completely duplicated on lines 186-194, containing "Next, we carry out gene ontology for differentially ... control cells (Figure 3E)".
10. Line 280 TRRUST online software (www.grapedia.org) provides an incorrect URL. Additionally, what factors other than SP1 were found to regulate KRT14 and were these not considered?
11. Does the expression of SP1 change in control and Y641F cells?
12. Line 357 refers to Figure 6N-6P, but should be 7N-7P.
13. Line 360 refers to Figure (7R-6S), a typo, should be (7R-7S). Authors used a few select genes to compare direction of expression. A correlation plot with coefficient of KRT14 and EZH2 in samples (or tool, e.g., cbioportal.org) should be provided so the reader can understand degree of correlation in samples.
14. The term "hyperactivated" or "hyperactive" to describe histone methylation is confusing (especially since H3K27me3 is a repressive mark); catalytic EZH2 can be hyperactive, but the mark H3K27me3 is simply increased or elevated.
15. Typo: "Kevin et.al" -> "Kevin et al."

Responses to the reviewers' comments

We heartily thank the reviewers for their scrupulous and enthusiastic evaluation of our manuscript. We also appreciate their constructive criticisms, responding to which has helped us improve our manuscript manifolds. New additions/changes are highlighted in blue in the revised manuscript. Given below is our point-by-point response to their concerns:

Responses to the comments of reviewer 1.

REVIEWER COMMENTS

Reviewer #1, expert in EZH2 and breast cancer metastasis (Remarks to the Author):

This is an interesting study on the importance EZH2 enzymatic activity on TNBC metastasis and a novel mechanism. There are a few comments that need to be addressed.

1. The investigators compare a SET deletion mutant (truncated EZH2) with a point mutation Y-641-F to make strong conclusions. The dSET mutant, as it is a deletion mutant, may have unexpected functions that are not solely due to the lack of enzymatic activity. Another approach is necessary to validate the results obtained with dSET. Also, while an activating mutation of EZH2 is a good approach, at least another approach to enhance EZH2 H3K27me3 activity should be used to validate the results.

Response: We express our sincere gratitude to this reviewer for critically reviewing our manuscript and coming up with very positive comments. As suggested by the reviewer, we have added new data in Figure 1L-1T (Total 9 panels) to support our findings.

In addition to activation mutation of EZH2 approach, here we achieved selective gain of function of H3K27me3 without altering EZH2 level by knocking down UTX or KDM6A (New Figure 1L) in TNBC cells and assessed their invasive potential in comparison to control (New Figure 1M, 1N). New data show that UTX KD results in a selective increase in H3K27me3 level along with marked enhancement in TNBC invasive potential.

As per reviewer's recommendations, we took another approach to further validate Δ SET associated results. We made TNBC cells where EZH2 SET domain was swapped with a less catalytically active EZH1 SET domain and assessed their migratory potential in comparison to control and WT EZH2 overexpression. The data were added as new Figure 1O, 1P, 1Q, which demonstrate that weakening of catalytic function of EZH2 causes reduction of TNBC invasive capabilities.

Further, in another approach for selective loss of H3K27me3 without altering the EZH2 level, we used the EZH2 pharmacological inhibitor drug EPZ6438 and demonstrated that EPZ6438 is highly selective in inhibiting the catalytic function of EZH2 and it significantly diminished TNBC invasive potential as is documented in new Figure 1R, 1S, 1T. The *in vivo* impact of EPZ6438 in inhibiting TNBC metastasis has already been documented in Figure 9N - 9O in the revised manuscript.

2. The *in vitro* studies in Figure 1 only compare Y-641-F to control, no experiments with dSET or EZH2 enzymatic activity inhibition are shown.

Response: We thank the reviewer for this excellent suggestion, bringing more clarity in our findings. In the revised manuscript, we have added dSET data in all the missing panels (Figure 1H, 1I, 1J, 1K) of the previous version of Figure 1.

3. 4T1 cells are aggressive breast cancer cells with already high levels of EZH2 expression. May not be the best model to further increase EZH2 activity.

Response: 4T1 is an aggressive cell line, but further induction of EZH2 activity via full-length EZH2 overexpression or selective H3K27me3 increase by Y641 mutant construct transfection demonstrate a significant increase in migration and invasion phenotype in both *in vitro* and *in vivo* conditions, vehemently suggesting that the system is definitely not saturated at the basal level. Further, we have tested several basal-like TNBC cell lines like HCC1806 and MDA-MB-468 (used in the manuscript to validate our results in human counterparts) in the orthotopic xenograft model to study metastasis. However, none of the cells developed spontaneous metastasis in nude and NOD-SCID mice, orthotopic xenograft models, even after prolonging the experiments up to 180 days. Therefore, we restricted our *in vivo* study to 4T1 cells only.

4. Further explanation on how genes were “manually selected” for validation is needed.

Response: We are really grateful, for the reviewer brought forward a very valid point and in response to that we have revamped the RNA- seq data, analysis and validation. New data has been added in Figure 3E-3I of revised Figure 3.

In summary, we did RNA seq analysis again in biological triplicates of each group and performed GSEA analysis to segregate the top 50 up and down-regulated genes, from which we selected 16 genes that correlated with migration and invasion phenotype. After stringent validation in individual Control vs. Y641 samples as well as primary tumor vs. lung, liver, splenic metastatic samples, we came up with KRT14 being the only gene that qualifies, considering both the criteria mentioned above.

5. There are defined targets of EZH2 in breast and other cancers (e.g. DKK1, hox genes). Were these genes present in this study?

Response: *DKK1* gene is not mapped in our study, but *hox* genes are present in our RNA-seq analysis. *hoxC6*, *hoxd3* and *hoxb5os* are downregulated in the Y641 group as compared to the control. However, none of them have notched a spot in our top 100 genes as inferred by our GSEA analysis.

6. In Figure 7, how are peritoneal metastasis defined? There are no histological images to show cancer cells in the peritoneum or in peritoneal fluid.

Response: We appreciate the reviewer for their astuteness. Now we have added IHC images of lung, liver and spleen metastasis as well as pictures of metastatic cells recovered from peritoneal fluid in the revised Figure 2H and Figure 2I.

7. The correlation shown between EZH2 and KRT14 is through data mining of transcript expression (7R). There is no demonstration that EZH2 and KRT14 proteins are concordantly expressed in breast cancer, and their association in TNBC and TNBC metastasis.

Response: We thank reviewer for this great suggestion and we firmly believe that the addition of such data will greatly enhance the quality of our manuscript. Now, we have not only added the EZH2 and KRT14 correlation plot (New Figure 8C) but we've also added IHC data of human TNBC patient samples to demonstrate EZH2, H3K27me3 and KRT14 protein expression in TNBC primary tumors and their respective metastatic counterparts (New Figure 8D-8G).

8. EZH2 KD has been shown previously to reduce invasion, migration, and other tumorigenic functions, and needs to be referenced appropriately (Moore HM. Breast Cancer Res Treat 2013 138:741-52, Anwar T Nat Commun 2018 9:2801)

Response: We apologise for this mistake. Respective suggested references have been added in the revised manuscript (Reference no 14 and 16)

In addition to figures mentioned above, Results, Materials Methods and Figure legend sections have been modified according to the above observations in the revised manuscript and have been highlighted in blue.

Reviewer #2, expert in breast cancer metastasis (Remarks to the Author):

In this MS, the authors investigated the underlying mechanism of basal-like TNBC peritoneal metastasis. They found EZH2 catalyzed H3K27me3 modification of KRT14 promoter region prevented the binding of SP1 transcriptional factor and therefore relieved the transcriptional repression of KRT14. Inhibition of EZH2 or KRT14 significantly blocked tumor metastasis and increase overall survival of experimental animals.

In general, this MS provided a potential therapeutic target for TNBC peritoneal metastasis, which is a rare and challenging clinical presentation. However, the main conclusion that EZH2-mediated H3K27me3 promoted breast cancer metastasis was not new (Nat Commun. 2018 Jun 29;9(1):2547. Nat Commun. 2018 Jul 18;9(1):2801.). Besides, lack of clinical data to support the conclusion will weaken the data solidarity. Detailed comments list below.

Major comments,

1. The authors should collect clinical samples to observe the correlation between EZH2-H3K27me3-KRT14 level and tumor distant metastasis, especially on sites of peritoneum, and abdominal organs, such as liver and spleen.

Response: First of all, we thank the reviewer for the positive remarks regarding our manuscript. We agree that the reviewer's first comment is critical in fortifying our manuscript but nonetheless, it is equally challenging too. Peritoneal metastasis has an abysmal prognosis and median patient survival is less than 6-12 months. Most of the patients die even before diagnosis. Further, MRI and CT scans are the first line of diagnostic choices than needle biopsy in case of peritoneal metastasis. Therefore, collecting clinical samples for peritoneal metastasis has huge limitations, particularly in TNBC patients. Still, as per reviewer's suggestion, we put in our diligent efforts and managed 10 TNBC samples, that have matched primary tumors and distant metastases.

Our detailed screening methodology and inclusion/exclusion criteria for the above 10 matched samples is as follows:

A total of 3500 Breast Cancer patients reported to RGCIRC, India between the years 2015-2019 were screened. The time frame chosen was such that a 2 year follow up period could be accounted for. Out of these 3500 patients, 600 cases qualified as triple-negative BC patients based on IHC and Her2 evaluation by FISH. They had also undergone surgery and other treatments at RGCIRC. Complete follow-up data of these 600 patients was analysed to look for disease progression in the form of local recurrence or distant metastasis. Approximately 60 patients had recurrence at varying time points. Of these 60 cases, 10 TNBC cases that had a matched tumor block and a matched metastatic site block were selected.

- a) TNBC cases which were metastatic and could not be included for the study were those where the metastatic site was inaccessible for a biopsy and the PET scan imaging suggested that it was metastatic, but a repeat biopsy was not done.
- b) A large number of cases had depleted blocks, and blocks not containing enough representative tissue for IHC.

The details of these 10 TNBC samples are enlisted in Supplementary Table S6 and IHC staining for EZH2, H3K27me3 and KRT14 in parallel sections of primary tumors and matched Lung, Lymph Node and Liver metastasis are shown in New Figure 9D. Quantitative analysis of EZH2, H3K27me3 and KRT14 expression in primary tumors versus matched metastasis has been documented in New Figure 9E-9G.

2. In the metastasis models, the authors used 4T-1 (Y641-F) and HCC1806 cells. Why HCC1806 cells were not transfected with Y641-F? Does HCC1806 cells have higher H3K27me3 modification?

Response: We have only used Y641-F in 4T1 cells for our *in vivo* metastasis model but not HCC1806 cells because orthotopic HCC1806 inoculation in nude and NOD-SCID mice does not metastasize to secondary organs. Even when we extend our experiments up to 180 days, HCC1806 only forms primary tumors in NOD-SCID mice. Therefore, we were resorted to restrict our metastasis studies to 4T1.

3. In the animal experiments, the authors should examine ki67, H3K27me3-KRT14 expression (IF blotted as figure 3N) both in primary tumors and metastases to confirm the in vitro results.

Response: As per the reviewer's suggestion, we have studied ki67, H3K27me3-KRT14 expression in cells isolated from primary tumors versus metastatic spleen through confocal microscopy. We have observed that H3K27me3 and KRT14 are highly overexpressed in splenic metastatic cells as compared to cells isolated from primary tumor (Figure 4J). However, expression of Ki-67 remains unaffected in both the conditions, further concreting our *in vitro* observations (Figure 4K).

Minor comments,

1. What did "control" group mean, cells overexpressed with wt EZH2 or empty vector? In figure 1B, 3H the expression of EZH2 didn't changed a lot in Y641-F group compared to control, had control group also been transfected with similar amount of EZH2?

Response: Control cells were transfected with an equal amount of plasmid containing the same backbone as the EZH2Y641-F plasmid but lacking an insert, thereby serving as empty vector. The avant-garde study by Damian *et al.*, (*Blood*. 2011 Feb 24;117(8):2451-9.2011) reported that the overexpression of EZH2 Y641-F mutant acts dominantly to enhance the H3K27me3 level, therefore, the expression of EZH2 did not change robustly in the Y641-F group compared to the control.

2. In figure 1C-E, we can see Y641-F significantly decreased tumor volume, however, there were no significance marks. At the same time, in sup figure 1, it seems Y641-F group had the similar tumor volume with the other two groups. These results were not consistent. The authors should check the data and explain the results.

Response: We thank the reviewer for pointing out this mistake. We have now re-analysed the data and fixed the discrepancies, in the revised Figure 1C-1D and Supplementary Figure 1.

3. In figure 1G, it's not clear why all 4T1 Y641-F OE tumors had metastasis to the contralateral body. How to explain the route of metastasis?

Response: Due to early-stage dorsal imaging of mice, it appeared that 4T1 Y641-F OE tumors had metastasis to the contralateral body. In the revised Figure 1F, we have added ventral images of mice just below the existing panel where the robust peritoneal metastasis can be clearly seen in Y641-F group.

4. In figure 4B-E, the authors better provide ChIP-seq results to show the broad peaks of H3K27me3 binding in the upstream promoter region of KRT14. So we can easily tell the enrichment region of H3K27me3 in the promoter, and how Y641-F affect H3K27me3 intensity.

Response: We outrightly agree with the reviewer's suggestion regarding ChIP-seq. It is imperative we mention here that in 2019, we had already made an attempt to perform RNA seq and ChIP seq in the same samples, unfortunately the Indian companies failed to deliver the ChIP-seq data, though they provided the RNA-Seq data. We had to cancel their ChIP-Seq orders (Proofs attached at end of the response letter). After this futile attempt, we decided to validate our results by performing extensive individual ChIP assays as documented in Figure 5 and Figure 6, to strengthen our conclusion.

5. To show the correlation between EZH2 and KRT14 expression, the authors better provide the correlation analysis.

Response: Correlation analysis has been provided in the new Figure 8C.

6. In figure 2E, 3N, 6K, the authors should provide scale bars.

Response: Now, we have provided scale bars in figure 2E, 3N (4I, revised version), 6K (7K, revised version) in the revised Figures.

7. In line 357, it should be Figure 7N-7P; and in line 361, it should be Figure 7R-7S.

Response: We are extremely sorry for this mistake, it has now been rectified in the revised manuscript.

In addition to figures mentioned above, Results, Materials Methods and Figure legend sections have been modified according to the above observations in the revised manuscript and have been highlighted in blue.

Reviewer #3, expert in breast cancer transcriptomics (Remarks to the Author):

“H3K27me3 mediated KRT14 upregulation promotes TNBC peritoneal metastasis” by Verma et al. uses in vitro and in vivo modeling to show that splenic metastasis of triple-negative breast cancer (TNBC) is governed by elevated (“hyperactivated”) H3K27me3 via the increased catalytic activity of EZH2. EZH2 has been characterized as an oncogene and is overexpressed in many different cancer types. It is a catalytic subunit of PRC2 and functions with EED and SUZ12 as a histone methyltransferase to silence active transcription through deposition of H3K27me3. Verma et al. focus on the catalytic function of EZH2 in basal-like TNBC, although the authors note this has already been described in part by several publications (including a recent study of the Mittal lab, Yomtoubian et al., 2020). The novelty of this work comes from the identification of KRT14, a known marker of TNBC, to be transcriptionally activated by EZH2. Although the therapeutic potential of EZH2 inhibitors in TNBC has been proposed previously (Yomtoubian et al., 2020), the mechanism of action described here is different. Overall, this is an interesting study that finds a role for increased H3K27me3 in peritoneal metastasis in TNBC.

Summary: The authors demonstrate that increased (“hyperactive”) H3K27me3 promotes TNBC migration in mice using in vivo and in vitro studies. To do this the authors engineered 4T-1 stable cells with catalytically inactive EZH2 (NC-EZH2) or catalytically hyperactive Y641F EZH2 protein. These genetically modified 4T-1 and control cells were implanted subcutaneously and orthotopically into mice. While NC-EZH2 tumor sizes did not change relative to control, Y641F tumors appeared smaller and Y641F tumor bearing mice lost significant body weight compared to control. Live animal imaging was used to assess the metastatic potential of groups of cells and found that Y641F mice had significant increase in metastasis compared to control. A wound healing assay and trans-well chamber confirmed Y641F mutant cells had higher migratory and invasive potential than WT.

The pattern of metastatic spread was assessed by fluorescence imaging of harvested organs. Control cells tended to spread to lungs whereas, Y641 cells tended to colocalize to the spleen and liver. Splenic cells had highest expression of H3K27me3. Animal imaging of inoculated splenic metastatic cells versus primary tumor cells show splenic cells result in profound increase of TNBC peritoneal metastasis.

RNA-sequencing identified KRT14 among others as significantly upregulated in Y641F cells. Earlier reports corroborate KRT14 as a marker of TNBC basal subtype. Here the authors validated KRT14 mRNA expression and protein levels in Y641F cells and show KRT14 expression is upregulated in Y641F compared to control and to cells treated with tazemetostat an EZH2 inhibitor. To investigate whether H3K27me3 can positively regulate KRT14 they perform ChIP assay and find reduction of H3K27me3 marks, but increased H3K4me3 marks around KRT14 promoter. The authors investigate SP1 as a transcriptional regulator of KRT14, identify its binding motif in the promoter of KRT14 and test association

with knockdown. They suggest H3K27me3 may compact the GC box region in the KRT14 promoter to inhibit SP1 binding and allow transcription of KRT14. EZH2 knockdown found reduced peritoneal metastasis compared to control and increased survival in mice. Though an FDA approved EZH2 inhibitor tazemetostat for sarcomas reduced TNBC peritoneal metastasis, the primary tumor burden was the same compared to control.

Several points need clarification:

1. Figures 1C and 1E should show the statistical significance of comparisons in tumor volume and weight, respectively.

Response: First of all, we extend our deepest, heartfelt gratitude to this reviewer for thoroughly reading and summarising our research findings and coming up with such valuable acumen, embodiment of which would magnificently upgrade the stature of our manuscript. As per the reviewer's recommendations, we have now added statistical significance of comparisons in tumor volume and weight, respectively, in revised Figure 1C and 1D (1E, older version).

2. The Figure 3B volcano plot shows significant differential expression analysis in just 284 genes (92 up, and 192 down at $p < 0.05$ and $fc \geq 2$). There are few genes here; a table of the gene expression and differential statistics between control and Y641F in the RNA-sequencing data should be provided. Are the p-values in DE analysis FDR-corrected?

Response: We are really thankful to the reviewer for the expert opinion that actually prompted us to recast the whole RNA-seq experiments. We re-performed RNA-seq in three biological replicate samples of Control vs. Y641 TNBC cells and re-analysed the data as per reviewer's recommendations considering the p value in DE analysis having FDR correction. We did GSEA analysis to segregate top 50 up and down-regulated genes. A total of 100 top-ranked, up and down-regulated genes were selected to explore their role in cancer metastasis. Based on intensive literature contemplation, out of 100 genes, a total of 16 genes were screened that showed metastatic potential, specifically in breast and other cancers (Supplementary Table 1). After stringent validation in individual Control versus Y641 samples as well as primary tumor versus lung, liver, splenic metastasis samples, we came up with KRT14, which is the only gene that has qualified considering both the aforementioned criteria. The newly analysed data has been shown in revised Figure 3E to 3I.

3. Figure S2A should be sorted according to significance similarly to Figure 2B.

Response: As per reviewer's recommendation, GSEA was implemented by repeating RNA sequencing with three biological replicates in control and Y641-F cells. Since the GSEA analysis is a more unbiased approach for RNA seq analysis, therefore, we deleted figures S2A and S2B from the manuscript.

4. Caption for Figure S2A is KEGG pathway analysis of candidate genes and indicates ($p < 0.5$) was considered for "statically significantly difference". There are typos in the quotes and is the written p-value cutoff ($p < 0.5$) accurate?

Response: As we have completely revised the RNA-seq data to comply with the reviewer's recommendation, we had to remove Figure S2A from the revised manuscript.

5. On page 7 (line 174) it is stated "we extensively searched literature for metastasis related genes..." A little more detail how these genes were 'selected'.

Response: As per reviewer's suggestion, we performed GSEA analysis to segregate the top 50 up and down regulated genes. A total of 100 top-ranked up and down-regulated genes were selected to explore their role in cancer metastasis. Based on literature perusal (Supplementary Table 1), out of 100 genes, 16 genes that showed metastatic potential specifically in breast and other cancers were selected for validation.

6. Figure 3C heatmap appears to have red as low and blue high. The opposite is more common (red high, blue low), style issue.

Response: We thank reviewer for this suggestion. Now, we have revised the figure as per their recommendation.

7. Neither S2A nor S2B appear to show significant enrichment of metastasis related pathways, yet these were specifically identified in the manuscript (line 176 and heatmap Figure 3C).

Response: The supplementary Figures S2A and S2B have been removed from the revised manuscript as we have metamorphosed the RNA seq data as per the reviewer's suggestion.

8. Line 179 is unclear and should be reworded: "...differentially expressed genes to identify the up and down regulated metastasis related KEGG pathways and most of the manually selected genes, we found common in the metastasis associated pathways. Additionally, the authors should clarify what are "most of the manually selected genes". A pathway analysis using an unbiased approach, such as. GSEA that includes expression of all genes between control and Y641F would clarify this analysis.

Response: We again performed RNA-seq in biological triplicates of control and Y641 group and analysed data as per reviewer's suggestion. Performed unbiased GSEA analysis to segregate top 50 up and down regulated genes, from which 16 genes that are found correlated with migration and invasion phenotype were selected. After stringent validation in individual Control vs Y641 samples as well as primary tumor vs lung, liver, splenic metastasis samples, we came up with KRT14, which is the only gene that qualifies, considering both the aforementioned criteria.

9. There are 9 lines 178-186 that are completed duplicated on lines 186-194, containing "Next, we carry out gene ontology for differentially ... control cells (Figure 3E)".

Response: We are extremely thankful to the reviewer for pointing out this error, the same has been modified in the revised manuscript.

10. Line 280 TRRUST online software (www.grapedia.org) provides an incorrect URL. Additionally, what factors other than SP1 were found to regulate KRT14 and were these not considered?

Response: We are sorry for the inconvenience caused; the corrected URL has been placed. Multiple analyses via diverse publicly available softwares, (Eukaryotic promoter data base, JASPER

and TRRUST) and intense literature perusal, converged our quest to SP1 as the major regulator of KRT14.

11. Does the expression of SP1 change in control and Y641F cells?

Response: We analysed SP1 expression in control and y641F cells and presented data in Figure 6E of revised manuscript. We did not find any change in SP1 expression of two different groups.

12. Line 357 refers to Figure 6N-6P, but should be 7N-7P.

Response: We apologise for the mistake and the figure has been shuffled in the revised manuscript and corrected as Figure 9N-9P.

13. Line 360 refers to Figure (7R-6S), a typo, should be (7R-7S). Authors used a few select genes to compare direction of expression. A correlation plot with coefficient of KRT14 and EZH2 in samples (or tool, e.g., cbiportal.org) should be provided so the reader can understand degree of correlation in samples.

Response: EZH2 and KRT14 correlation plot has been incorporated in new Figure 8C.

14. The term "hyperactivated" or "hyperactive" to describe histone methylation is confusing (especially since H3K27me3 is a repressive mark); catalytic EZH2 can be hyperactive, but the mark H3K27me3 is simply increased or elevated.

Response: We agree with the reviewer's opinion and now we have replaced the term 'hyperactive H3K27me3' and revised our manuscript accordingly.

15. Typo: "Kevin et.al" -> "Kevin et al."

Response: We apologise for the mistake. The correction has been done in the revised manuscript.

In addition to figures mentioned above, Results, Materials Methods and Figure legend sections have been modified according to the above observations in the revised manuscript and have been highlighted in blue.

In our pursuit, we have solemnly tried to address all the comments put forward by the reviewers. We are highly indebted for their keen insight and worthy comments, which have helped us enhance the calibre of our work. We look forward to their positive response.

Email**Dipak Datta**

Cancellation of PO against indent # 59196

From : Dipak Datta
<dipak.datta@cdri.res.in>

Fri, Dec 20, 2019 12:53 PM

 1 attachment**Subject :** Cancellation of PO against indent
59196**To :** SPO CDRI <spo@cdri.res.in>,
Rizvan Khan
<rizvan.khancsir@gmail.com>**Cc :** Sanjeev Meena
<meena726@gmail.com>, ayushi
verma <ayushicdri@gmail.com>

Dear SPO Sab,

Please cancel the PO against indent # 59196 as vendor was unable to deliver the product. Please find attached the PO.

Thanks and Regards,
Dipak

--

Dipak Datta, Ph.D
Principal Scientist
Cancer Biology Division
Life-Science South, Lab # 006
CSIR-Central Drug Research Institute
B.S. 10/1, Sector-10, Jankipuram Extension
Sitapur Road, Lucknow-226031, INDIA

59196 Datta PO Cancellation.pdf
91 KB

सी.एस.आई.आर.-केन्द्रीय औषधि अनुसंधान संस्थान

(वैज्ञानिक तथा औद्योगिक अनुसंधान परिषद्)

बी.एस. 10/1, सेक्टर 10, जानकीपुरम विस्तार, सीतपुर रोड, पोस्ट बॉक्स नं. 173, लखनऊ - 226021 (भारत)

CSIR-Central Drug Research Institute

(Council of Scientific & Industrial Research)

B.S. 10/1, Sector 10, Jankipuram Extension, Sitapur Road, Post Box No.: 173, Lucknow- 226021 (India)

Phone : (0522) 2771940, 2771960, 2961202, Fax : 91- (522) 2771941

Gram : CENDRUG, Web : <http://www.cdriindia.org>

Purchase Order No: 2019-20/IND59196PO1 /Rijwan

Date 26/07/2019

From

The Director
Central Drug Research Institute
LUCKNOW

To

Gene Print Lifesciences Pvt. Ltd.
D-1/253,
Sector F
Jankipuram
Lucknow
India
226021

Reference: Your Offer/Quotation No: **GPLS/CDRI/2018-19/103**

Dated: **27/06/2019**

Dear Sirs

Please arrange to supply the articles noted below strictly on the terms and conditions stated on the reverse of this order latest by 4-6 WEEKS failing which the order may be treated as cancelled.

S.No	Item Name	Catalog No	Price	Qty	Total (INR)
1	Validation of Library & Generation of 75 Long raw sequence minimum 20-30 million reads. 1	GPNGS-CHIP	20000.0	6	120000.0
Warranty :					AS PER QUOTATION
Agency Comm:					0.0
Total after Discounts:					120000.0
Excise Duty (18.0) % :					21600.0
Total after Taxes:					141600.0
Grand Total :(INR)					141600.0

Remarks :

Delivery Schedule :

4-6 WEEKS

Delivery Terms

NORMAL :

Note :

SPO Comments:

OK

TERMS & CONDITIONS FOR ORDERS

1. The order should be acknowledged by return of post confirming the acceptance of rates and all other conditions of supply. Any variation in price/specifications must be intimated before effecting the supply.
2. The Number and Date of this order must be quoted in the bill and in all correspondence relating to this supply.
3. Additional charges such as Packing & Forwarding, Sales Tax/GST, Insurance, Postal/Courier, octroi etc. will not be paid unless specifically mentioned in order and supported by payees? Cash Money receipts or original vouchers.
4. (a) Payment should be claimed through pre-receipted bills drawn in duplicate, original copy being signed over a Revenue Stamp (Amount exceeding Rs.5000/- INR)
5.
 1. Payment will be made on receipt of materials in good conditions after Inspection by crossed Account-Payee Cheque on State Bank of India(Name of the Bank)/CDRI Branch at Lucknow.
 2. Installation of Equipment must be completed within 30 days of delivery & report submitted to SPO, CDRI
6. All damaged or unapproved goods shall be returned at your cost & risk and the incidental expenses incurred thereon shall be recoverable from any of your bills.
7. Printed conditions, if any, sent along with the quotation shall not be binding on us.
8. The date of delivery should be strictly adhered to failing which the purchase order is liable to be cancelled.
9. As time is the essence of this order, the date of delivery should be strictly adhered to, otherwise the Director, CDRI, Lucknow (Name of the Lab) reserves the right not to accept the delivery in part or in full and to claim liquidated damages @ 1% per week subject to a maximum of 10% of the total value of supply order.
10. Local firms are requested to deliver the goods in our Stores before 4.00 P.M. on any working day.
11. The equipment should be guaranteed against any manufacturing defect for a period of atleast 12 months from the date of successful installation. In case any part or whole of the equipment is found to be defective during the guarantee period, then the same will have to be replaced/repared free of cost at our premises(FOR CDRI Stores)
12. Any dispute arising out of this contract shall be subject to the decision of the courts situated at Lucknow, India only.
13. **NO PARTIAL/PART SHIPMENT**

STORES & PURCHASE OFFICER

Phone:91-522-2618367
 91-522-2629504
 Fax:91-522-2629504
 e-mail: spo@cdri.res.in

A-TERMS OF SUPPLY : FOR CDRI/Ex- Godown ..Stores...
 B-PAYMENT TERMS: BILL BASIS
 C- DELIVERY TIME: 4-6 WEEKS
Forwarding Agent:

Note: Local Suppliers to note that the material should be delivered to our STORE SECTION and not to any other Section or Individual.

Reviewer comments, second round review -

Reviewer #2 (Remarks to the Author):

In this revised manuscript, the authors had well addressed all the critiques raised by the reviewers by providing several new data panels and literature review. The manuscript is now suitable for publication in the NC journal.

Reviewer #3 (Remarks to the Author):

H3K27me3 mediated KRT14 upregulation promotes TNBC peritoneal metastasis by Ayushi Verma, et al. uses in vitro and in vivo modeling to show splenic metastasis of triple-negative breast cancer (TNBC) is governed by hyperactivated H3K27me3.

Most previous points were addressed in the revision. However there are still issues with the gene expression and GSEA analysis that need to be fixed.

1. After reading the Materials and methods section, it appears differential analysis was performed using EdgeR. Differential expression defaults are assumed but should be stated clearly.

2. In the revision, GSEA is used in multiple places, but the description is hard to parse:

- In their response "Reviewer 3" number 8: "We again performed RNA-seq in biological triplicates of control and Y641 group and analysed data as per reviewer's suggestion. Performed unbiased GSEA analysis to segregate top 50 up and down regulated genes, from which 16 genes that are found correlated with migration and invasion phenotype were selected."

- Additionally, on line 210 the authors identified 2142 differentially expressed genes, then from lines 211-217: "...Gene Set Enrichment Analysis (GSEA) was performed to evaluate the characteristic dynamic pattern between control and Y641-F cells. The transcriptome profile of two experimental settings in GSEA analysis provides a heat map of the top 50 upregulated and down-regulated genes (Figure 3F). A total of 100 top-ranked upregulated and down-regulated genes were selected to explore their role in cancer metastasis further. Based on the literature, out of these 100 genes, 16 genes were screened that showed metastatic potential, specifically in breast and other cancers (Supplementary Table S1)."

- Also, in their response "Reviewer 3" number 5. "As per reviewer's suggestion, we performed GSEA analysis to segregate the top 50 up and down regulated genes."

Note that GSEA evaluates a gene set with respect to a fully ranked gene list (generally genes are ranked by differential expression) to identify whether the gene set is enriched at the top or bottom of the list; the GSEA "leading edge" test statistic depends on the ranks of the genes in the gene set within the overall list.

- Can the authors please clarify the use of GSEA to segregate genes?

- Are the authors actually using GSEA or just ranking genes from differential expression analysis, and if the latter please specify criteria?

3. According to their manuscript GSEA was used with Hallmark signatures (line 636).

- Which signatures were most significant?

- GSEA is mentioned throughout but there is no evidence to support GSEA was used as no signatures are shown.

Minor points:

- Fig 3F: Heatmap legend labels "high" and "low"; if these are z-scored fold-changes provide them.

- Fig 3G: Legend indicates these are metastasis-related genes from GSEA, but no GSEA is ever shown. It appears to be a 'select' group of genes from Figure 3F.

- Small detail but the color scale bar (presumably indicating log₂FC, not labeled) is not centered.

Reviewer #4 (Remarks to the Author):

The manuscript reveals that EZH2 promotes TNBC peritoneal metastasis potentially through a mechanism involving H3K27me₃ mediated KRT14 upregulation. The author further indicates that “our preclinical findings posit a rational insight for the clinical development of H3K27me₃ inhibitors like tazemetostat as a targeted therapy against TNBC”

1. However, the key findings of EZH2 inhibition could inhibit TNBC dissemination and metastasis have already been described by Yomtoubian et al (<https://doi.org/10.1016/j.celrep.2019.12.056>). Furthermore, Granit et al (<https://doi.org/10.1038/onc.2012.390>) has already demonstrated that EZH2 inhibition leads to a decreased KRT14+ population. The novelty and significance of this study is only modest.

2. Although there is a newly proposed mechanism involving H3K27Me₃ activated KRT14 gene expression, the mechanism or KRT14 itself is not sufficiently support EZH2 mediated peritoneal metastasis, particularly, how H3K27Me₃-mediated suppression versus H3K27Me₃-mediated activation of gene expression is regulated, and how other known EZH2-regulated genes involved in breast cancer coordinately work with KRT14 in the big picture of cancer metastasis.

3. Furthermore, EZH2-mediated gene activation has been recently uncovered and robustly studied. High impact papers have revealed that the EZH2 binds cMyc and p300 through a cryptic transactivation domain to mediate gene activation, where occupancy of the activation histone modifications, H3K27ac and H3K9ac are enriched, but lacks H3K27me₃ (<https://doi.org/10.1038/s41556-022-00850-x>; <https://doi.org/10.1073/pnas.1914866117>). Therefore, the authors need to discuss these findings, and EZH2-cMyc-p300 and H3K27ac/H3K9ac on KRT14 promoter should be examined.

4. Also notably, many clinical and preclinical studies (<https://doi.org/10.1080/10428194.2018.1430795>; <https://doi.org/10.1038/s41467-022-30105-0>; <https://doi.org/10.1126/scitranslmed.aaz5387>) have indicated that treatment EZH2 methyltransferase inhibitor alone does not provide preclinical/clinical benefits for solid tumors including breast cancers. The authors should discuss these findings and significantly tone down the statement like “targeting the methyltransferase activity of EZH2 could be a top-notch strategies to prevent TNBC metastasis in the clinic” on page p5.

5. The authors did not provide adequate response/data to address several important questions raised by the reviewers. For example:

Question #3 by reviewer 1-- There are other xenograft mouse models and genetic mouse models (e.g. tail vein injected MDA-MB-231 in NOD/SCID mouse, PyMT mouse model) can be used for studying breast cancer metastasis. One 4T1 xenograft model is not sufficient.

Question #4 by reviewer 2 and Question #4 by reviewer 1-- The CHIP-seq data is crucial to demonstrate H3K27Me3 is indeed the key player mediating peritoneal metastasis by revealing the identities of the H3K27Me3 activating gene targets across the genome. Manually selection of one KRT14 gene is not adequately supportive of the findings.

Response to the Reviewers' comments

We are grateful to the reviewers for spending their precious time in evaluating our manuscript and come up with constructive criticisms, responding to which has helped us improve our manuscript. New additions/changes are highlighted in blue in the revised manuscript. Given below is our point-by-point response to their concerns:

Responses to the comments of reviewer 2

REVIEWER COMMENTS

Reviewer #2 (Remarks to the Author):

In this revised manuscript, the authors had well addressed all the critiques raised by the reviewers by providing several new data panels and literature review. The manuscript is now suitable for publication in the NC journal.

Response: We sincerely thank the reviewer for accepting the revised version of the manuscript and recommended the same for publication in *Nature Communications*.

Reviewer #3 (Remarks to the Author):

H3K27me3 mediated KRT14 upregulation promotes TNBC peritoneal metastasis by Ayushi Verma, et al. uses in vitro and in vivo modelling to show splenic metastasis of triple-negative breast cancer (TNBC) is governed by hyperactivated H3K27me3.

Most previous points were addressed in the revision. However, there are still issues with the gene expression and GSEA analysis that need to be fixed.

Overall Response: First of all, we thank reviewer for the positive remarks on our earlier revised version. We are really grateful for the reviewer's expert opinion on our RNA-seq data analysis and we strongly believe that further addressing reviewer's points will definitely improve the clarity of our manuscript. We are really sorry for overemphasizing GSEA analysis in our last rebuttal letter and sincerely apologise to the reviewer if it created any confusion. As per your suggestion, we performed GSEA that offered a heat map of 100 top ranked differentially expressed genes between control and experimental groups. As rightly pointed out by the reviewer, we did not opt for classical GSEA representation, however, top 100 ranked genes were considered for further analysis according to our observed phenotype. Now, we have thoroughly revised our manuscript (Material and Methods, Results, Figure, Figure Legend, Supplementary Table and References) to bring more clarity on transcriptome analysis.

1. After reading the Materials and methods section, it appears differential analysis was performed using EdgeR. Differential expression defaults are assumed but should be stated clearly.

Response: We are sorry for the unintentional error. Now, we have revised our manuscript according the reviewer's recommendation (Page No. 26, Revised Material Method Section).

2. In the revision, GSEA is used in multiple places, but the description is hard to parse:

- In their response "Reviewer 3" number 8: "We again performed RNA-seq in biological triplicates of control and Y641 group and analysed data as per reviewer's suggestion. Performed unbiased GSEA analysis to segregate top 50 up and down regulated genes, from which 16 genes that are found correlated with migration and invasion phenotype were selected."

- Additionally, on line 210 the authors identified 2142 differentially expressed genes, then from lines 211-217: "Gene Set Enrichment Analysis (GSEA) was performed to evaluate the characteristic dynamic pattern between control and Y641-F cells. The transcriptome profile of two experimental settings in GSEA analysis provides a heat map of the top 50 upregulated and down-regulated genes (Figure 3F). A total of 100 top-ranked upregulated and down-regulated genes were selected to explore their role in cancer metastasis further. Based on the literature, out of these 100 genes, 16 genes were screened that showed metastatic potential, specifically in breast and other cancers (Supplementary Table S1)."

- Also, in their response "Reviewer 3" number 5. "As per reviewer's suggestion, we performed GSEA analysis to segregate the top 50 up and down regulated genes."

Note that GSEA evaluates a gene set with respect to a fully ranked gene list (generally genes are ranked by differential expression) to identify whether the gene set is enriched at the top or bottom of the list; the GSEA "leading edge" test statistic depends on the ranks of the genes in the gene set within the overall list.

- Can the authors please clarify the use of GSEA to segregate genes?

- Are the authors actually using GSEA or just ranking genes from differential expression analysis, and if the latter please specify criteria?

Response: Yes, we are fully agreeing with the reviewer's suggestions that GSEA evaluates a gene set with respect to fully ranked gene list to identify whether the gene is enriched at the top or bottom of the list; GSEA leading edge test statistics depends on the rank of the genes in the gene set within the overall list. As mentioned in our overall response, instead of classical GSEA representation, we have used GSEA that provides 100 top ranked differentially expressed genes between control and experimental groups. Considering the reviewer's comments, now, we have revised relevant result section (Page No. 9), Figure Legend of 3F, Supplementary Table-1, to make it more easily understandable.

3. According to their manuscript GSEA was used with Hallmark signatures (line 636).

- Which signatures were most significant?

- GSEA is mentioned throughout but there is no evidence to support GSEA was used as no signatures are shown.

Response: As we have not shown the classical GSEA representation, instead we used the same to focus on 100 top ranked differentially expressed genes between control and

experimental groups. We have revised the Results section (Page No. 9) and Figure Legend (Figure 3F) (Page No. 43) of current version of the manuscript accordingly and toned down the emphasis on GSEA analysis.

Minor points:

- *Fig 3F: Heatmap legend labels "high" and "low"; if these are z-scored fold-changes provide them.*

Response: Thank you for your comment; GSEA 4.2.2, by default, does not provide a scale bar for the heat map of the ranked genes. Actually, genes were ranked based on the z-score as we have chosen signal-to-noise ranking metric for the GSEA analysis. So, we have removed the heatmap color scale in the Revised Figure 3F and made required changes to the respective Figure Legend (Page No. 43-44) of the revised manuscript to avoid any confusion.

- *Fig 3G: Legend indicates these are metastasis-related genes from GSEA, but no GSEA is ever shown. It appears to be a 'select' group of genes from Figure 3F.*

Response: Now, we have modified the relevant Figure and respective Figure Legend (Figure 3G) to make it easier to understand.

- *Small detail but the color scale bar (presumably indicating log2FC, not labeled) is not centered.*

Response: As suggested by the reviewer, colour scale bar has been updated in the revised Figure 3G.

Reviewer #4 (Remarks to the Author):

The manuscript reveals that EZH2 promotes TNBC peritoneal metastasis potentially through a mechanism involving H3K27me3 mediated KRT14 upregulation. The author further indicates that “our preclinical findings posit a rational insight for the clinical development of H3K27me3 inhibitors like tazemetostat as a targeted therapy against TNBC”

1. However, the key findings of EZH2 inhibition could inhibit TNBC dissemination and metastasis has already been described by Yomtoubian et al (<https://doi.org/10.1016/j.celrep.2019.12.056>). Furthermore, Granit et al (<https://doi.org/10.1038/onc.2012.390>) has already demonstrated that EZH2 inhibition leads to a decreased KRT14+ population. The novelty and significance of this study is only modest.

Response: We are fully aware of the Yomtoubian et al., (<https://doi.org/10.1016/j.celrep.2019.12.056>) study (Reference No. 19 of current revised

version) that was referred by the reviewer and we have thoroughly discussed the significance of their observations and limitations even in the first version of the manuscript submitted to the journal (Page No. 4 and 17). Their major finding (catalytic function of EZH2 restricts TNBC metastasis) is an immense support of our observations in spite of considering the pro-tumorigenic functions of EZH2 protein. Yomtoubian et.al, published their studies in 2020, but we had two abstracts published in 2018 (*Verma et al., Cancer Medicine 7, 30-30; 2018*) and AACR 2020 (*Verma et al., Cancer Research 80, 16_Supplement, 3837-3837; 2020*) (Listed as Reference No. 22 and 23 of current revised version) clearly advocate the fact that both the studies are independently carried out and especially journals like *Nature Communications* do support reproducible science. Further, we have demonstrated how catalytic function of EZH2 can change the landscape of TNBC metastasis and deciphered completely novel mechanisms for the same in a more physiologically relevant mammary fat pad orthotopic metastatic model, instead of direct inoculation of breast cancer cells into systemic circulation which recapitulates only metastatic colonization and circumvent the primary disease. In such cases, metastatic colonization is strongly influenced by the site of cell inoculation (*Gengenbacher, et.al, Nature Reviews Cancer volume 17, pages751–765; 2017,*) which is actually the major limitations of Yomtoubian et al., study.

In the context of Granit et al (<https://doi.org/10.1038/onc.2012.390>) studies, we sincerely thank reviewer for pointing out a very relevant paper in support of our observation and now, we have cited (Reference No 49) the particular paper and discussed the significance of their findings in the revised manuscript (Revised Discussion Section, Page 19). However, in Granit *et al.*, manuscript, authors just demonstrated EZH2 knock-down results in KRT14 downregulation but there was no experimental evidence supporting whether it was due to loss of catalytic function of EZH2 or reduction of EZH2 protein. In our manuscript, we not only demonstrated how catalytic function of EZH2 (H3K27me3) regulates KRT14 expression by adapting multiple approaches (**Figure -3, Figure- 4, Figure -5 and Figure -6**) but also put forward evidence in clinical samples where H3K27me3 and KRT14 expression are positively correlated in human TNBC metastasis (**Figure-8**).

2. Although there is a newly proposed mechanism involving H3K27Me3 activated KRT14 gene expression, the mechanism or KRT14 itself is not sufficiently support EZH2 mediated peritoneal metastasis, particularly, how H3K27Me3-mediated suppression versus H3K27Me3-mediated activation of gene expression is regulated, and how other known EZH2-regulated genes involved in breast cancer coordinately work with KRT14 in the big picture of cancer metastasis.

Response: The primary goal of this manuscript is to dissect the role of EZH2 protein (non-catalytic EZH2) vs H3K27me3 (catalytic EZH2) in TNBC growth and metastasis. In course of deciphering their role, we discovered that catalytic function of EZH2 not only promotes metastasis but also changes the metastatic landscape of TNBC. Understanding the molecular mechanisms of this process through unbiased approaches like RNA sequencing and further

validation in basal TNBC metastatic state as well as multiple genetically modified *in-vitro* and *in-vivo* systems (**Figure-3, Figure-4, Figure-5 and Figure-7**), we came to a conclusion that H3K27me3 dependent KRT14 regulation is one of the crucial factors for regulating TNBC peritoneal metastasis.

We completely acknowledge the reviewer's opinion regarding the possibility of H3K27Me3-mediated suppression versus H3K27Me3-mediated activation of gene expression may be critical in delineating the whole process. Clearly suggesting the above fact, now we have added relevant discussion point (Page No. 20) in the revised manuscript. Though studies are on-going, but we strongly believe that detailed examination of all genes is beyond the scope of the current manuscript and further additions will dilute the focus of the current manuscript.

3. Furthermore, EZH2-mediated gene activation has been recently uncovered and robustly studied. High impact papers have revealed that the EZH2 binds cMyc and p300 through a cryptic transactivation domain to mediate gene activation, where occupancy of the activation histone modifications, H3K27ac and H3K9ac are enriched, but lacks H3K27me3 (<https://doi.org/10.1038/s41556-022-00850-x>;<https://doi.org/10.1073/pnas.1914866117>). Therefore, the authors need to discuss these findings, and EZH2-cMyc-p300 and H3K27ac/H3K9ac on KRT14 promoter should be examined.

Response: We understand the reviewer's concern regarding the role of EZH2 protein (Non-catalytic function) in modulating tumorigenesis. Undeniably, the linkage of EZH2 function in the context of tumorigenesis has been found quite versatile and cancer-type specific. As per the reviewer's recommendations, the above mentioned studies are cited (Reference No 17, 18) and discussed in the relevant places (Introduction: Page No. 3, Results: Page No. 12 and Discussion: Page No. 19) in the revised manuscript.

The reviewer might have missed but we have already shown in **Supplementary Figure-3** that there was no enrichment of EZH2 in the KRT14 promoter in 4T-1 and HCC-1806 cells. While we agree that the suggested experiment regarding EZH2-cMyc-p300 would be supportive if we get the enrichment of EZH2 in the KRT14 promoter in the first place. Therefore, the possibility of H3K27ac and H3K9ac occupancy in the KRT14 promoter via EZH2 does not arise in our experimental set up as we did not find EZH2 enrichment in the promoter of *KRT14* gene. These results are quite surprising to us but such instances are present in the literature (*Laugesen et.al., Mol Cell. 2019 Apr 4;74(1):8-18*). On the other hand, to confirm KRT14 activation, by performing ChIP assays, we had already shown

enrichment of well-known activation mark H3K4me3 in the KRT14 promoter in the **Figure-5** of main manuscript. For the reviewer's convenience and understanding, we are showing the EZH2 and H3K4me3 ChIP results below in addition to the documentation in the main manuscript.

The EZH2 enrichment analysis by ChIP-q PCR at KRT14 promoter

ChIP was performed in 4T-1 mouse and HCC1806 human using anti-EZH2 and IgG antibodies and then examined by real-time q-PCR, using respective primers of KRT14 gene. ChIP q-PCR results showing differential fold change in EZH2 enrichment at the promoter of KRT14 gene. (A and B) The analysis of enrichment for EZH2 in the KRT14 Promoter was performed at -0.2 Kb, -0.5 Kb, -1.1 Kb and +2.4 Kb regions from TSS respectively. Data points are average duplicate readings of samples; error bars, \pm S.D.

H3K4me3 ChIP analysis on KRT14 promoter

(A) ChIP was performed in 4T-1 mouse cells using anti-H3K4me3 and IgG antibodies and then examined by real-time q-PCR using primer pairs targeting -1.1 Kb to +2.4 Kb of the KRT14 gene (B) Same procedure for HCC1806 human cells as in G. (C) the differential fold change enrichment for H3K4me3 +2.4 Kb region from TSS in control and EZH2-Y641 4T1 cells by ChIP q-PCR. (D) The differential fold change enrichment for H3K4me3 was analysed in +2.4 Kb region from TSS in the HCC1806 cells treated either with vehicle control or 10 μ M EPZ6438.

4. Also notably, many clinical and preclinical studies (<https://doi.org/10.1080/10428194.2018.1430795>; <https://doi.org/10.1038/s41467-022-30105-0>; <https://doi.org/10.1126/scitranslmed.aaz5387>) have indicated that treatment EZH2 methyltransferase inhibitor alone does not provide preclinical/clinical benefits for solid tumors including breast cancers. The authors should discuss these findings and significantly tone down the statement like “targeting the methyltransferase activity of EZH2 could be a top-notch-strategies to prevent TNBC metastasis in the clinic” on page p5.

Response: Agreeing with the reviewer’s opinion, we have now removed “*top-notch strategies*” from the above-mentioned statements and toned down all such statements present in Introduction, Results and Discussion section of the manuscript and revised accordingly to the reviewer’s suggestion. Also, we have cited (Reference No 41, 42, and 63-65) and discussed findings of all the above studies in the Discussion section (Page No. 18, 20-21) of the revised manuscript.

5. The authors did not provide adequate response/data to address several important questions raised by the reviewers. For example:

Question #3 by reviewer 1-- There are other xenograft mouse models and genetic mouse models (e.g. tail vein injected MDA-MB-231 in NOD/SCID mouse, PyMT mouse model) can be used for studying breast cancer metastasis. One 4T1 xenograft model is not sufficient

Response: We agree with the reviewer's concern, however, as mentioned before and documented in several studies (as cited in Reference No. 43, 44, and 45), tail vein injection is not an ideal model for understanding organ-specific metastasis of breast cancer since the cancer cells directly spread into the blood circulation and seamlessly find shelter within lungs even though the cells do not possess metastatic properties. In this case, metastatic colonization largely depends on site of cancer cell inoculation, instead of representing true metastatic cascade. So, deciphering TNBC metastatic landscape, tail vein inoculation is not a suitable option. Further, TNBC is a highly heterogeneous group of cancers, where intrinsic subtype analysis suggests approximately 80% of TNBC belongs to the basal-like category (as cited in Reference No. 4), therefore, we majorly focus our study on basal-like TNBC and selected the TNBC cells with basal characteristics such as 4T-1, HCC1806 and MDAMB468. As also mentioned in the last rebuttal, besides, 4T-1 orthotopic model, we also tried to

develop the orthotopic TNBC xenograft models using HCC1806 and MDA-MB-468 cells in nude and NOD/SCID mice. Both the cell lines were found to develop primary tumors in the mammary fat pad but unable to metastasize to the secondary organs even after prolonging the experiments up to 180 days. Moreover, aligning with the current reviewer’s suggestion, before attempting to develop orthotopic TNBC xenograft model using MDAMB231 cells, we first checked the impact of H3K27me3 inhibition on KRT14 expression in MDAMB231 cells. Unfortunately, we did not observe KRT14 downregulation upon EPZ6438 treatment even though the H3K27me3 level was markedly inhibited. Similar treatment resulted in marked KRT14 reduction in all the other basal like TNBC cells. Mesenchymal origin of MDAMB231 could be the contributory factor for this discrepant result. Due to the lack of correlation between H3K27me3 and KRT14, we did not perform the xenograft study with MDAMB231 cells. In the rebuttal we are showing the western blot results in EPZ6438 treated MDAMB231 cells along with 4T1, HCC1806, and MDAMB468 cells. In addition, it is noteworthy to mention that beyond preclinical validation in animal models, we showed positive correlation between H3K27me3 and KRT14 expression in TNBC patient samples where multiple TNBC metastatic cases including peritoneal or liver metastasis represents significant association of these two further void the requirement of validation of our finding in additional preclinical xenograft experiments.

Western blot analysis for KRT14 expression in MDAMB231, 4T-1, HCC1806 and MDAMB468 after EPZ6438 treatment.

(A) The MDAMB231 cells were either treated for vehicle or 5 μ M and 10 μ M EPZ6438 to check the expression of EZH2, H3K27me3, KRT14 and β -actin by immunoblot. (B and C) 4T-1, HCC1806 and MDAMB468 cells were either treated for vehicle or 10 μ M EPZ6438 to check the expression of EZH2, H3K27me3, KRT14 and β -actin by immunoblot.

Question #4 by reviewer 2 and Question #4 by reviewer 1-- The ChIP-seq data is crucial to demonstrate H3K27Me3 is indeed the key player mediating peritoneal metastasis by revealing the identities of the H3K27Me3 activating gene targets across the genome. Manually selection of one KRT14 gene is not adequately supportive of the finding.

Response: First of all, requirement of ChIP-seq experiment was originally suggested by the Reviewer # 2 as one of the ‘Minor comments’ but it was not raised by Reviewer #1. We would like to reiterate that in the last rebuttal, we expressed our views and provided evidence of our honest effort to perform the same and Reviewer # 2 was convinced with our response and accepted our article for publication in the journal. Regarding the selection of KRT14 gene, we would like to convey that KRT14 lead came from our unbiased RNA-seq analysis and further validation through multiple modalities, instead of manual selection of the same as mentioned by the reviewer. Following RNA-seq (**Figure-3**), GSEA analysis ranked top 100 genes in which 16 of them correlated with migration, metastasis phenotype and out of 16, KRT14 is the only gene which qualified at basal state of TNBC peritoneal metastasis where we observed maximum H3K27me3 and KRT14 expression (**Figure-3**). Finally, we validated H3K27me3-KRT14 axis extensively in *in-vitro*, *in-vivo* and TNBC patient samples focusing on metastasis.

With our honest effort, we made an attempt to address all the concerns of the reviewers. We are highly grateful for their thoughtful insight and useful comments, which have helped us to enhance the quality of our work. We look forward to their positive response.

Reviewer comments, third round review -

Reviewer #3 (Remarks to the Author):

The authors have adequately addressed prior critiques.

Reviewer #4 (Remarks to the Author):

The authors have not fully addressed questions #2-#4. There are concerns on the mechanism involved in EZH2/H3K27me3-mediated KRT14 activation, particularly whether the proposed mechanism only governs KRT14 activation, given that peritoneal metastasis is a complex biological process that seemingly requires coordinated changes of a cohort of genes.

While EZH2 is not enriched in KRT14 promoter, other potential mechanisms that could directly enhance H3K27me3 and H3K4me3, such as lost UTX/JMJD3 or enriched MLL, are not investigated. Without comprehensive ChIP-seq data, it is difficult to determine whether this is a stand-alone mechanism specific for KRT14 gene and the scope/importance of these mechanistic findings could not be better appreciated.

Moreover, since many clinical and preclinical studies have indicated that EZH2 methyltransferase inhibitor treatment alone does not provide preclinical/clinical benefits for breast cancer, it is not clear how this treatment could succeed as a potential therapeutic option for TNBC metastatic patients albeit good xenograft treatment data. The authors discussed the literatures but did not elaborate how these discrepancies were caused.

Response to the Reviewers' comments

We are thankful to the reviewers for devoting their precious time in evaluating our manuscript and come up with useful comments, responding to which has helped us improve our manuscript. New additions/changes are highlighted in blue in the revised manuscript. Given below is our point-by-point response to their concerns:

REVIEWER COMMENTS

Reviewer #3 (Remarks to the Author):

The authors have adequately addressed prior critiques.

Response: We sincerely thank the reviewer for accepting the revised version of the manuscript.

Reviewer #4 (Remarks to the Author):

The authors have not fully addressed questions #2-#4. There are concerns on the mechanism involved in EZH2/H3K27me3-mediated KRT14 activation, particularly whether the proposed mechanism only governs KRT14 activation, given that peritoneal metastasis is a complex biological process that seemingly requires coordinated changes of a cohort of genes.

Response: As stated in the previous rebuttal letter, we completely agree with the fact that TNBC peritoneal metastasis is multifactorial and can be governed by several factors. In our manuscript, we deciphered EZH2/H3K27me3-KRT14 linkage is one of the crucial factors in modulating TNBC peritoneal metastasis. Follow up studies are on-going but further addition of other modulatory factors will dilute the focus of the current manuscript. Keeping these limitations of our studies in mind, now we have further toned down our claims and amended even the Abstract in revised version of the manuscript.

While EZH2 is not enriched in KRT14 promoter, other potential mechanisms that could directly enhance H3K27me3 and H3K4me3, such as lost UTX/JMJD3 or enriched MLL, are not investigated. Without comprehensive ChIP-seq data, it is difficult to determine whether this is a stand-alone mechanism specific for KRT14 gene and the scope/importance of these mechanistic findings could not be better appreciated.

Response: In the last version, reviewer suggested EZH2 protein mediated *cMyc* and *p300* recruitment in the promoter of KRT14 could be the critical modulators to look at, without noticing that there was no recruitment of EZH2 in the promoter of KRT14 as demonstrated in the original manuscript. When we pointed out the same and rule out the possibility *cMyc* and *p300* experiments, reviewer came up with the new hypothesis for the involvement of UTX/JMJD3 or enriched MLL for KRT14 regulation. The purpose of the ChIP experiment with EZH2 is to see if EZH2 can function as a transactivator (PRC2-independent function) in the KRT14 promoter. Although we did not find EZH2 enrichment in the KRT14 promoter that does not imply that other proteins can enhance H3K27me3 level instead of EZH2. As stated earlier, many ChIP-seq studies have shown that discernible PRC2 binding occurs in a relatively small fraction of the genome, whilst its catalytic products are found on 70%-80% of all histone H3 despite the fact PRC2 being responsible

for all H3K27 methylation (PMID: 20150217, PMID: 30951652). These interactions are most likely transient, with very short residence times, and their "ChIP invisibility" may also reflect the fact that they occur with minimally involved interaction surfaces, such as the PRC2 catalytic site engages with the histone H3 tail, which may further impair efficient antibody cross-link capture. Notably, single-molecule microscopy studies of PRC2 dynamics also confirm that 80% of PRC2 is highly mobile, while the remaining PRC2 is stably bound with a long residence time in histone tails (PMID: 29891558 and PMID: 30951652). These unforeseen findings indicate that stable PRC2 binding is not required for the formation of high global levels of H3K27me3 (PMID: 30951652). In support of the above literature, we have also observed H3K27me3 mediated KRT14 mRNA upregulation without the recruitment of EZH2 in the promoter of KRT14.

Though we have done some UTX loss of function experiments in modulating H3K27me3 expression to fulfil the comments of the other reviewer as an accessory strategy, primary focus of our manuscript is to understand how EZH2 catalytic function (H3K27me3) drives TNBC growth and metastasis. In response to the reviewer's comment and concern for non EZH2 driven H3K27me3 alterations in modulating KRT14 expression, now we have modified the title of the revised paper and made it clear that our focus is on EZH2 mediated H3K27me3 changes. Therefore, further addition of reviewer recommended experiments will dilute the focus of our current manuscript.

As stated in the first rebuttal letter, in response to one of the minor comments of Reviewer #2, we completely understand the significance of ChIP-seq in our studies. But it is imperative we mention here that in 2019, we had already made an attempt to perform RNA seq and ChIP seq in the same samples, unfortunately the Indian companies failed to deliver the ChIPseq data, though they provided the RNA-Seq data. We had to cancel their ChIP-Seq orders (Proofs attached in the earlier version). After this futile attempt, we decided to validate our results by performing extensive individual ChIP assays as documented in Figure 5 and Figure 6, to strengthen our conclusion. However, we never claim that this is a stand-alone mechanism specific for KRT14 gene; rather same can be true in other cases that would even increase the impact of our studies.

Moreover, since many clinical and preclinical studies have indicated that EZH2 methyltransferase inhibitor treatment alone does not provide preclinical/clinical benefits for breast cancer, it is not clear how this treatment could succeed as a potential therapeutic option for TNBC metastatic patients albeit good xenograft treatment data. The authors discussed the literatures but did not elaborate how these discrepancies were caused.

Response: As per the reviewer's recommendations, now we have elaborated and discussed the possible reasons for the above-mentioned discrepancies in the revised version of the manuscript. (Discussion Section, Page No. 20-21)

With our truthful effort, we made our best effort to address all the concerns of the reviewer. We are thankful for their insight and valuable comments, which have helped us to improve the quality of our work. We look forward to their positive response.